# Loss of the lysosomal protein CLN3 triggers c-Abl-dependent YAP1 pro-apoptotic signaling

Neuza Domingues [ID][1✉], Alessia Calcagni' [ID][2,3], Sofia Freire [ID][1], Joana Pires[1], Ricardo Casqueiro [ID][1], Ivan L Salazar [ID][1], Niculin Joachim Herz[4,5], Tuong Huynh [ID][4,5], Katarzyna Wieciorek[6], Tiago Fleming Outeiro [ID][6,7,8,9], Henrique Girão [ID][10], Ira Milosevic [ID][1,11], Andrea Ballabio [ID][2,3,4,5,12] & Nuno Raimundo [ID][1,13,14✉]

## Abstract

**Batten disease is characterized by early-onset blindness, juvenile dementia and death within the second decade of life. The most common genetic cause are mutations in CLN3, encoding a lysosomal protein. Currently, no therapies targeting disease progression are available, largely because its molecular mechanisms remain poorly understood. To understand how CLN3 loss affects cellular signaling, we generated human CLN3 knock-out cells (CLN3-KO) and performed RNA-seq analysis. Our multi-dimensional analysis reveals the transcriptional regulator YAP1 as a key factor in remodeling the transcriptome in CLN3-KO cells. YAP1-mediated pro-apoptotic signaling is also increased as a consequence of CLN3 functional loss in retinal pigment epithelia cells, and in the hippocampus and thalamus of $Cln3^{\Delta7/8}$ mice, an established model of Batten disease. Loss of CLN3 leads to DNA damage, activating the kinase c-Abl which phosphorylates YAP1, stimulating its pro-apoptotic signaling. This novel molecular mechanism underlying the loss of CLN3 in mammalian cells and tissues may pave a way for novel c-Abl-centric therapeutic strategies to target Batten disease.**

**Keywords** Lysosomes; Batten Disease; Lysosome-Nucleus Communication; YAP1; DNA Damage
**Subject Categories** Autophagy & Cell Death; Molecular Biology of Disease; Neuroscience

## Introduction

Neuronal ceroid lipofuscinoses (NCLs), also collectively referred to as Batten disease, are the most prevalent neurodegenerative lysosomal storage diseases (LSDs) (Platt et al, 2018). The pathology of NCLs exhibits devastating manifestations such as blindness, seizures or epilepsy, dementia, loss of cognitive and motor skills, ultimately leading to death around 20 years of age (Simonati and Williams, 2022). There are currently no therapies targeting the progression of NCLs, mostly due to the lack of knowledge about the underlying disease mechanisms. The NCLs are characterized by mutations in 13 genes encoding different CLN proteins, e.g., CLN2, CLN3, and CLN5. The most common mutations are in the gene encoding CLN3 (Mole and Cotman, 2015), particularly a 1.02 kb deletion which removes exons 7 and 8 ($CLN3^{\Delta7/8}$). This mutation results in an open-reading frame disruption that leads to either mRNA decay or to translation of a protein which is truncated at the C-terminus (Centa et al, 2023; Miller et al, 2013).

CLN3 is an ubiquitously-expressed multi-pass transmembrane protein initially reported to be predominantly localized at the lysosomal membrane (Mirza et al, 2019) and, more recently, at the Golgi apparatus (Calcagni' et al, 2023). Additionally, CLN3 was shown to regulate the trafficking of the cation-independent mannose-6-phosphate receptor, revealing an important function in the secretion of lysosomal enzymes, and in autophagic-lysosomal reformation (Calcagni' et al, 2023). CLN3 was also shown to be involved in the export of glycerophosphodiesters (GPDs) from the lysosome (Laqtom et al, 2022). These metabolites are generated through lysosomal degradation of phospholipids, which are first metabolized into lysophospholipids, and then further converted into GPDs. Interestingly, brain tissue from mice lacking CLN3 revealed lysosomal accumulation of phospholipid catabolism products, including the degradation intermediate lysophosphatidylglycerol (Laqtom et al, 2022), which is due to the inhibitory effect of GPDs on other enzymes of the lysosomal phospholipid catabolism pathway (Nyame et al, 2024). Despite this implication of CLN3-loss-of function in lysosomal reformation and cellular lipid homeostasis, its broader effects on the crosstalk of lysosomes with

[1]Multidisciplinary Institute of Ageing, Centre for Innovative Biomedicine and Biotechnology (CIBB), University of Coimbra, Coimbra, Portugal. [2]Telethon Institute of Genetics and Medicine (TIGEM), Naples, Italy. [3]Department of Translational Medical Sciences, Federico II University, Naples, Italy. [4]Department of Molecular and Human Genetics, Baylor College of Medicine, Houston, TX, USA. [5]Jan and Dan Duncan Neurological Research Institute, Texas Children's Hospital, Houston, TX, USA. [6]University Medical Center Göttingen, Department of Experimental Neurodegeneration, Center for Biostructural Imaging of Neurodegeneration, Göttingen, Germany. [7]Translational and Clinical Research Institute, Newcastle University, Newcastle upon Tyne, UK. [8]Max Planck Institute for Multidisciplinary Sciences, Göttingen, Germany. [9]Deutsches Zentrum für Neurodegenerative Erkrankungen (DZNE), Göttingen, Germany. [10]Coimbra Institute for Clinical and Biomedical Research (iCBR), Centre for Innovative Biomedicine and Biotechnology, Academic and Clinical Center of Coimbra, Faculty of Medicine, University of Coimbra, Coimbra, Portugal. [11]Centre for Human Genetics, Nuffield Department of Medicine, University of Oxford, Oxford, UK. [12]SSM School for Advanced Studies, Federico II University, Naples, Italy. [13]Department of Cellular and Molecular Physiology, Penn State College of Medicine, Hershey, PA, USA. [14]Penn State Cancer Institute, Hershey, PA, USA. ✉E-mail: neuza.domingues@uc.pt; nuno.raimundo@psu.edu

other organelles and the subsequent impact on cellular signaling remain unclear.

YAP1 is an important regulator of cell proliferation and death, and is considered a canonical effector of the Hippo pathway (Zhao et al, 2010), which regulates YAP1 phosphorylation at S127. In addition to this regulation of YAP1 activity through the Hippo pathway, which is usually associated with cancer cells and/or hypoxia response, other kinases phosphorylate YAP1, regulating its localization, interaction with co-activators and transcriptional activity. Accordingly, in response to DNA damage, the tyrosine kinase c-Abl relocates to the nucleus and phosphorylates YAP1 on tyrorine-357 (Levy et al, 2008). This stimulates YAP1 interaction with tumor suppressor p73 and pro-apoptotic gene expression (Lapi et al, 2008; Strano et al, 2005). In addition, in the context of LSDs, YAP1 was also found to interact with TFEB, contributing to the accumulation of autophagosomes (Ikeda et al, 2021). The activation of a pro-apoptotic signaling pathway caused by impaired lipid trafficking between lysosomes and the nuclear envelope opens a new paradigm for inter-organelle communication and cellular signaling. Here, we demonstrate in human cell lines and in the brain of a mouse model of Batten disease that loss of CLN3 function causes DNA damage, which triggers a pro-apoptotic response mediated by the transcription factor YAP1, in a c-Abl-dependent and Hippo-independent manner.

## Results and discussion

### Loss of CLN3 in human cells triggers YAP1 signaling

To determine the signaling consequences of the loss of CLN3, we generated a CLN3 knockout in human embryonic kidney (HEK) 293T cells (henceforth CLN3-KO) with CRISPR-mediated deletion of ten nucleotides in the CLN3 gene (Fig. EV1A), which causes premature termination (Fig. EV1B). The protein levels of CLN3 were strongly reduced in CLN3-KO cells (Fig. EV1C). First, we confirmed the expected lysosomal phenotypes of CLN3 loss-of-function (Calcagni' et al, 2023) in this cell line by evaluating lysosomal morphology. Upon staining the cells with an anti-Lamp1 antibody to mark the lysosomal membrane, we observed that the size of the Lamp1-positive particles was robustly increased in CLN3-KO (Fig. EV1D), while the total number of lysosomes was similar in CLN3-KO and wildtype (Wt, Fig. EV1E). In agreement with this result, Lamp1 protein expression was unchanged in CLN3-KO cells (Fig. EV1F). To confirm that the increase in lysosomal size was a direct consequence of CLN3 loss rather than an adaptive mechanism, we re-expressed CLN3 (CLN3-GFP) in the CLN3-KO cells, where it localized to the lysosomes as expected (Golabek et al, 1999) (Fig. EV1G). Expression of CLN3-GFP was sufficient to rescue the "swollen" phenotype of CLN3-KO lysosomes (Fig. EV1H). We next investigated lysosomal function in CLN3-KO cells by assessing lysosomal proteolytic activity and lumenal pH. Cells were loaded with the pH-insensitive Alexa-Fluor 647 and the pH-sensitive FITC-labeled dextran, or with Magic Red, a substrate of cathepsin B, which functions as a readout of cathepsin B activity. Of note, bafilomycin A1 (BafA1) treatment was used as a positive control for non-acidic lysosomes. We observed that the CLN3-KO cells had lower cathepsin B activity and higher pH than the Wt cells (Fig. EV1I). Interestingly, despite

the observed impairment of lysosomal function in CLN3-KO, lysosomal biogenesis and autophagy were not upregulated as possible compensatory mechanisms. By assessing the TFEB intensity in the nucleus of Wt and CLN3-KO cells (Fig. EV1J) and LC3-II levels in the presence or absence of BafA1 (Fig. EV1K), we found a decrease in TFEB nuclear recruitment and a decrease in autophagic flux in CLN3-KO cells. These results are consistent with the lack of significant differences in Lamp1 levels and total lysosomal number per CLN3-KO cells when compared with Wt.

To determine the key consequences of CLN3-KO in an unbiased manner, we performed bulk RNA sequencing (Fig. 1A). We identified 2288 differentially expressed genes (DEGs), with 909 upregulated and 1379 downregulated in CLN3-KO (Fig. EV1L; Table EV1). We then analyzed the pathways predominantly enriched in the DEG list, and observed that the top pathways were related to cell proliferation (cell cycle, mTORC1 signaling), retinoic acid signaling, and DNA damage (Fig. 1B). Next, we determined which transcription factors were predicted to mediate the changes observed in the transcriptome. Only two transcription factors were predicted to be hyperactive in CLN3-KO cells, YAP1 and SMAD1 (Fig. 1C). The enrichment of YAP1 targets in the CLN3-KO DEG list was much greater than for SMAD1; henceforth, we focused on YAP1 for further mechanistic investigations.

### YAP1 pro-apoptotic signaling is increased both in CLN3-KO cells and the mouse brain

Considering that YAP1's intracellular location is tightly regulated as it shuffles between the nucleus and cytoplasm, with nuclear localization being necessary for its transcriptional activity, we first determined the intracellular distribution of YAP1. We observed a robust increase in YAP1 nuclear localization in CLN3-KO cells, assessed by confocal microscopy (Fig. 1D) and by immunoblot of YAP1 on nuclear extracts (Fig. EV2A), consistent with increased YAP1 activity in these cells.

YAP1 signaling predominantly affects the balance between cell proliferation and cell death (Piccolo et al, 2022). Notably, a key feature of Batten disease is the loss of neurons and retinal cells by apoptosis (Piccolo et al, 2022). Therefore, we hypothesized that YAP1 could be promoting pro-apoptotic signaling in CLN3-KO cells. We compared the expression levels of several YAP1 transcriptional targets that are involved in pro-apoptotic signaling (PUMA, TP53, TP73, BAX, and DR5), and observed that these transcripts were present at higher levels in CLN3-KO cells (Fig. 1E), supporting increased YAP1 transcriptional activity. To confirm that the YAP1-associated pro-apoptotic signaling is a direct consequence of CLN3 loss and not a long-term adaptation, we performed a transient knockdown of CLN3 in HEK293T cells, using two siRNA constructs to control for off-target effects. This resulted in a decrease of CLN3 transcripts (Fig. EV2B) and CLN3 protein levels (Fig. EV2C), as well as in "swollen" lysosomes (Fig. EV2D), similarly to the CLN3-KO cells. We observed an increase in the transcript levels of pro-apoptotic YAP1 targets in CLN3-silenced HEK293T cells (Fig. 1F), suggesting that YAP1-driven pro-apoptotic signaling is a direct consequence of CLN3 loss. To further test this premise, we re-expressed CLN3 in CLN3-KO cells, and observed a decrease in the expression levels of YAP1 transcriptional targets (Fig. 1G).

We then sought to test if loss of CLN3 also triggered pro-apoptotic signaling in a cell type directly involved in the pathology of Batten disease. To achieve this, we used ARPE19 cells, a human

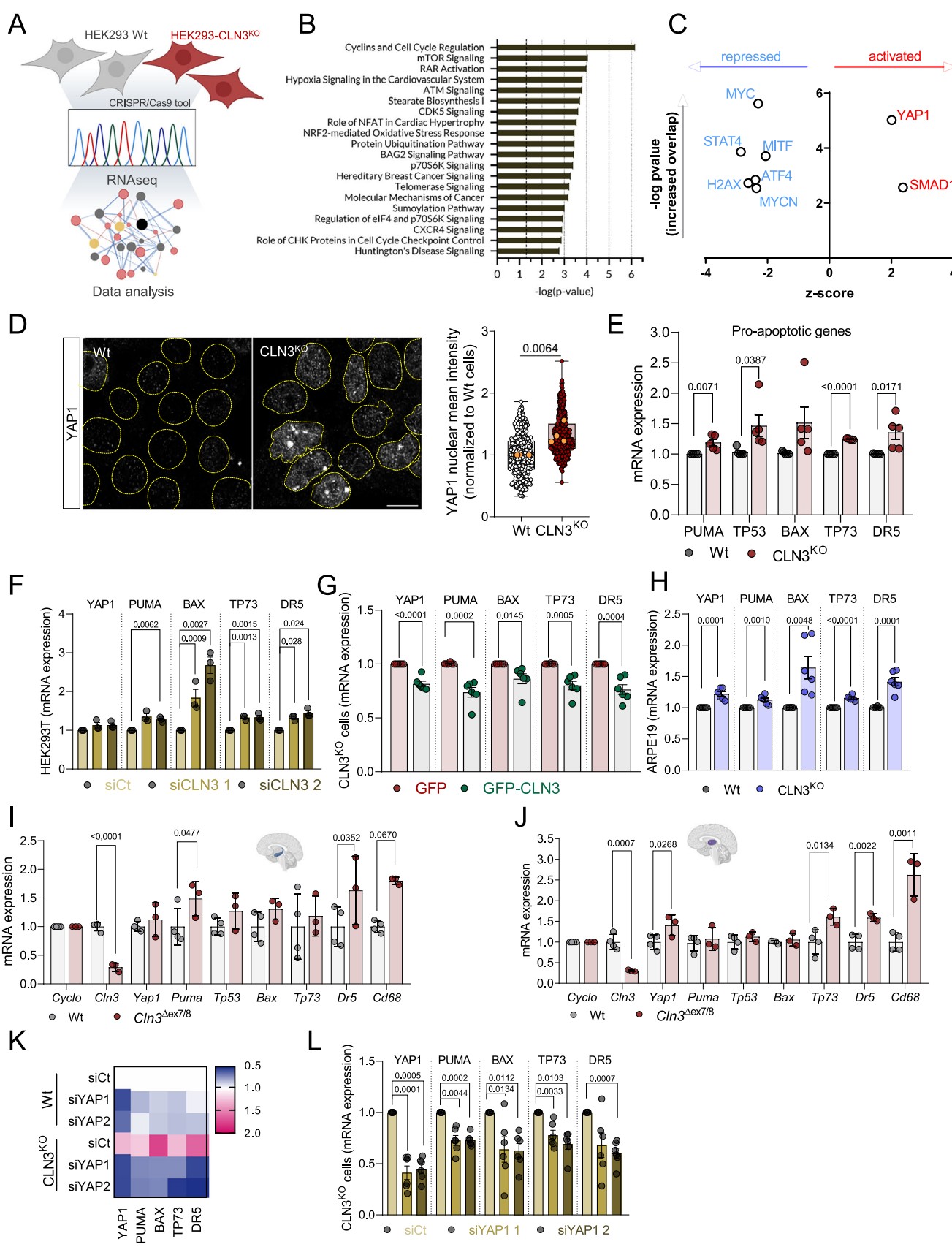

**Figure 1.   YAP1 signaling pathway is activated in CLN3^KO cells and drives a pro-apoptotic phenotype.**

(**A**) Schematic representation of the experimental design. (**B**) Pathway enrichment analysis of the differentially expressed genes (DEG). (**C**) Transcription factors predicted from the DEG list (**B**) to be either hyperactive (red) or repressed (blue) represented as a network. The transcription factor activity predictions and the interactions were determined using ingenuity pathway analysis (Qiagen). (**D**) Confocal fluorescence images of Wt and CLN3^KO cells immunostained for YAP1 protein. Nuclei are outlined by the yellow dashed lines using Hoechst staining. Scale bar 10 µm. On the right, quantification of the mean intensity of YAP1 at the nucleus of at least 100 cells per each condition. The orange dots represent the mean of each independent experiment ($n = 4$). Boxplots show 25th and 75th percentiles, median, and whiskers represent min to max values. (**E–H**) mRNA levels of YAP1 target genes involved in apoptosis in HEK293T Wt and CLN3^KO stable cell lines ($n = 5$) (**E**), in Wt cells with transient CLN3 depletion using two different siRNA ($n = 3$) (**F**), CLN3^KO cell line transfected with GFP or GFP-CLN3 ($n = 6$) (**G**) and in ARPE19 Wt and CLN3^KO stable cell line ($n = 6$) (**H**). (**I**, **J**) mRNA levels of YAP1-target pro-apoptotic genes in hippocampus (**I**) and thalamus (**J**) tissue collected from 12 months old control (Wt) (4 animals) and *Cln3^{Δ7/8}* animals (three animals). (**K**) Heat map from the expression level of pro-apoptotic genes from Wt and CLN3^KO cells transfected with siCt or two different siRNA against YAP1 ($n = 2$). (**L**) mRNA levels of YAP1 target genes involved in apoptosis in HEK293T CLN3^KO cell line transfected with control (siCt) or two different siRNA directed to the YAP1 transcript. With (**D**) exception, all the results are mean ± SEM. Statistical analysis with Tukey's post-test (**F**, **H**, **L**), and Student's unpaired *t*-tests when comparing two experimental groups (**D–J**). Right-tailed Fisher's exact test was applied to data from (**B**, **C**). Source data are available online for this figure.

model of retinal pigmental epithelium which degenerates in Batten disease (Puranam et al, 1997; Calcagni' et al, 2023). Our results show increased transcript levels of pro-apoptotic YAP1 targets in conditions of CLN3 loss-of-function in both ARPE19 stable CLN3-KO (Fig. 1H) or by transient CLN3 depletion (Fig. EV2E–H). This suggests that the transcriptional upregulation of pro-apoptotic genes in response to CLN3 loss is conserved in a cell type prominently affected in Batten disease.

We next determined whether the pro-apoptotic signaling resulting from the absence of CLN3 was also observed in vivo. For that, we employed the *Cln3^{Δ7/8}* mice and their Wt littermates, and analyzed the expression of YAP1 pro-apoptotic targets in two different brain regions, which have been found to progressively deteriorate in Batten disease patients, the hippocampus and thalamus (Hochstein et al, 2022). Importantly, we observed an increase in the transcript levels of pro-apoptotic YAP1 target genes: *Tp73, Dr5 and Cd68* in the hippocampus (Fig. 1I) and thalamus (Fig. 1J) of 12-month-old *Cln3^{Δ7/8}* mice compared to Wt animals. Additionally, to verify that the pro-apoptotic signaling triggered by loss of CLN3 was indeed mediated by YAP1, we transiently silenced YAP1 by siRNA in Wt and CLN3-KO cells (Fig. EV2I), and observed a significant decrease in the transcript levels of YAP1 pro-apoptotic targets in CLN3-KO cells, with a non-significant effect on Wt cells (Fig. 1K,L), indicating that the pro-apoptotic signaling observed in CLN3-KO cells is downstream of YAP1. Altogether, these results show that CLN3 depletion triggers a YAP1-mediated pro-apoptotic signaling pathway, in different human cell types as well as in vivo rodent models.

## YAP1-dependent pro-apoptotic signaling in CLN3-KO is mediated by its interaction with p73

YAP1 activity is regulated by post-translational modifications, which determine its intracellular localization and transcriptional activity. In particular, YAP1 pro-apoptotic activity has been associated with the phosphorylation of its tyrosine-357 (Y357) residue (Levy et al, 2008; Raghubir et al, 2021; Sugihara et al, 2018). Therefore, we tested whether phosphorylation of this residue (pYAP1^{Y357}) was affected in CLN3-KO cells. We observed an increase in pYAP1^{Y357} in whole cell extracts from HEK and ARPE19 CLN3-KO cells (Fig. 2A). Furthermore, we observed a robust enrichment of pYAP1^{Y357} in the nucleus of CLN3-KO cells (both types), whilst it was barely detected in the nucleus of Wt cells (Fig. 2B). Because our transcriptomic analysis revealed several

pathways related to DNA damage (Levy et al, 2008), we sought to test if induction of DNA damage would increase YAP1^{Y357} phosphorylation. Interestingly, despite increased pYAP1^{Y357} levels in HEK293T Wt cells upon doxorubicin-induced DNA damage, as previously described for cancer cells (Levy et al, 2008), no further increase was induced in CLN3-KO cells (Fig. EV3A). Additionally, the levels of YAP1 phosphorylated in serine127 residue (S127), which are associated with the activation of YAP1 by the Hippo pathway, and the consequent cytosolic retention of YAP1, were not changed in HEK293T CLN3-KO cells (Fig. 2C). To further confirm that Hippo signaling was not affecting pYAP1^{Y357}, we treated Wt and CLN3-KO HEK293T cells with Xmu-mp-1, an inhibitor of the MST1/2 kinases that regulate the phosphorylation of pYAP1^{S127} by Hippo signaling, and we found no effect on pYAP1^{Y357} (Fig. EV3B). Because Hippo-mediated pYAP1^{S127} can also be degraded by the proteasome, and thus affect the availability of YAP1, we treated Wt and CLN3-KO HEK293T cells with the proteasome inhibitor lactacystin, and found no impact on YAP1 levels or YAP1 phosphorylation at the Y357 residue (Fig. EV3C,D). Corroborating the lack of effect of the Hippo pathway on the activation of pYAP1^{Y357} in CLN3-KO cells, we subjected HEK293T Wt and CLN3-KO cells to hypoxic conditions (Fig. EV3E), known to deactivate Hippo signaling and thus to alleviate the inhibition of Hippo on YAP1, and observed that pYAP1^{Y357} levels were decreased in HEK293T Wt and CLN3-KO cells upon 24 h of hypoxia (Fig. EV3F). These results suggest that the activation of YAP1, and in particular the phosphorylation pYAP1^{Y357}, in CLN3-KO cells is unrelated to Hippo signaling.

To confirm that increased pYAP1^{Y357} is a specific consequence of CLN3-KO, we expressed CLN3-GFP in CLN3-KO cells, and observed a decrease in pYAP1^{Y357} (Fig. EV3G). Furthermore, pYAP1^{Y357} nuclear intensity was also reduced in CLN3-KO cells expressing CLN3-GFP (Fig. EV3H). To test if increased YAP1^{Y357} phosphorylation was also observed in vivo, we immunostained pYAP1^{Y357} in brain sections of *Cln3^{Δ7/8}* and Wt mice. Importantly, we found an increased number of pYAP^{Y357}-positive cells in the hippocampus and in the thalamus (Fig. 2D), as well as an increase in total levels of pYAP1^{Y357} in *Cln3^{Δ7/8}* brain tissue compared to Wt (Fig. 2E). Given the Batten's disease phenotype, we also verified if this finding is relevant for neuronal cells. Importantly, in primary murine neuronal cultures, increased levels of pYAP1^{Y357} were also observed when CLN3 was knocked down (Fig. EV3I). These results underscore the importance of YAP1^{Y357} phosphorylation in the activation of YAP1 in human CLN3-KO cells, in human CLN3-

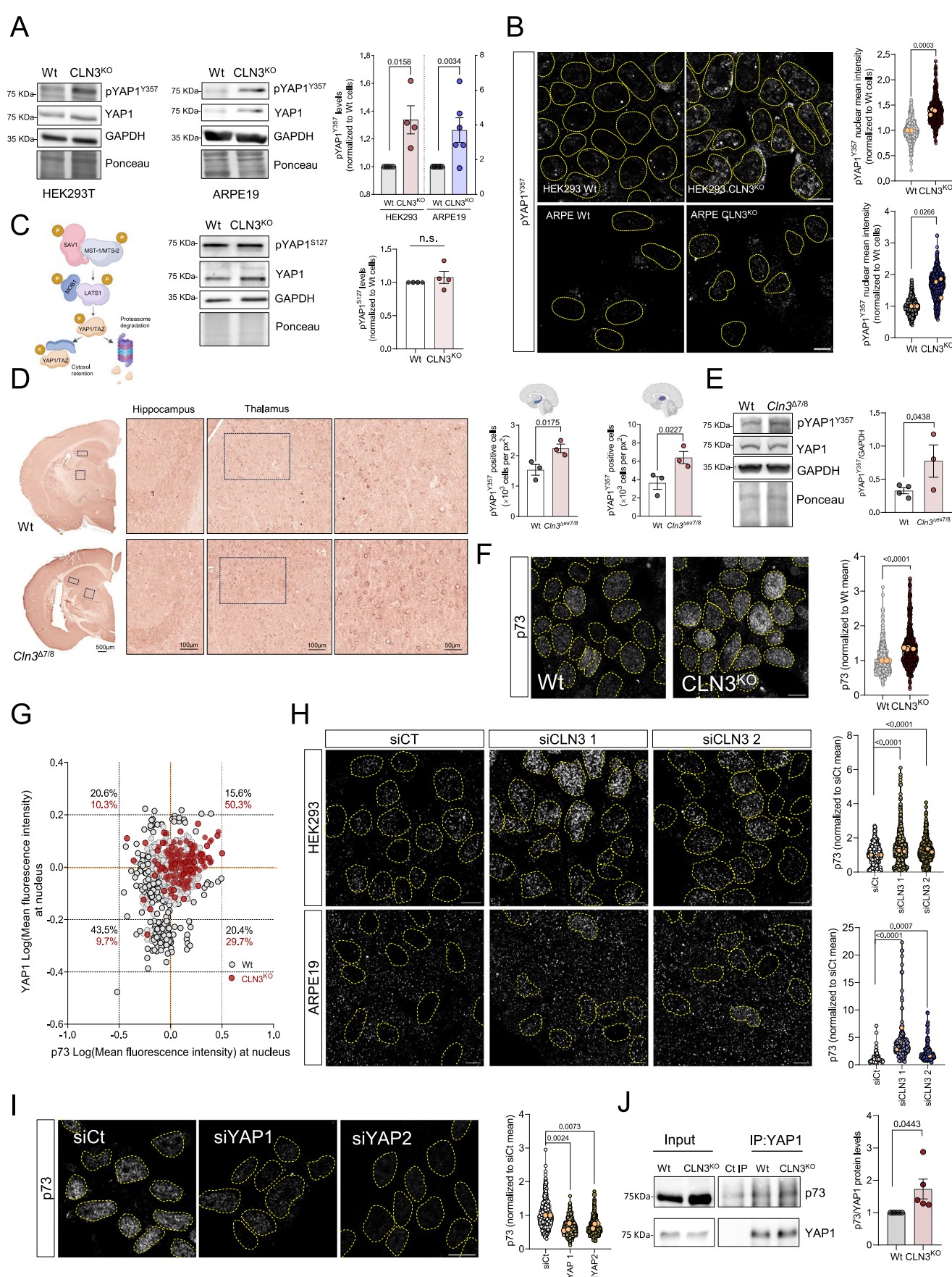

◀ **Figure 2. Loss of CLN3 promotes the interaction between YAP1 and p73.**

(A) Representative immunoblot image and quantification of YAP1 protein levels phosphorylated at the tyrosine residue 357 (pYAP1$^{Y357}$) in HEK293T ($n = 4$) and ARPE19 ($n = 6$) Wt and CLN3$^{KO}$ cells. GAPDH and Ponceau were used as loading controls. (B) Confocal fluorescence images of HEK293T ($n = 3$) and ARPE19 ($n = 3$) Wt and CLN3$^{KO}$ cells immunostained for pYAP1$^{Y357}$ protein. Nuclei are outlined by the yellow dashed line using Hoechst staining. Scale bar 10 μm. On the right, quantification of the mean intensity of YAP1 in the nucleus of at least 500 cells per each condition. (C) Schematic representation of the regulation of YAP1/TAZ activation in the Hippo pathway. SAV1 (protein salvador homolog 1) and MTS-1/MST-2 (mammalian Ste20-like serine/threonine kinases 1/2) complex phosphorylates and activates MOB1 (Mps one binder 1) and LATS1 (large tumor suppressor kinase 1), with consequent phosphorylation (P) of YAP1/TAZ complex, resulting in cytoplasmic retention and proteasome degradation. On the middle and left, representative immunoblot image and quantification of YAP1 protein levels phosphorylated at the serine residue 127 (pYAP1$^{S127}$) in HEK293T Wt and CLN3$^{KO}$ cells ($n = 4$). GAPDH and Ponceau were used as loading controls. (D) Immunohistochemistry analysis of pYAP1$^{Y357}$ staining of control (Wt) and *Cln3*$^{Δ7/8}$ animals. On the right, quantification of the number of positive nuclei to pYAP1$^{Y357}$ per the same area of hippocampus (first) and thalamus (second) ($n = 3$). (E) Representative immunoblot image and quantification of pYAP1$^{Y357}$ in Wt ($n = 4$) and *Cln3*$^{Δ7/8}$ animals ($n = 3$). GAPDH and Ponceau were used as loading controls. (F) Confocal fluorescence images of Wt and CLN3$^{KO}$ cells immunostained for p73 protein. Nuclei are outlined by the yellow dashed lines using Hoechst staining. Scale bar 10 μm. On the right, quantification of the mean intensity of p73 at the nucleus of at least 500 cells per each condition ($n = 4$). (G) Correlation between the intensity levels of nuclear YAP1 (x-axis) and p73 (y axis) proteins. At least 150 cells were analysed ($n = 2$). (H) Confocal fluorescence images and quantification of HEK293T (upper panels and graph, $n = 3$) and ARPE19 (bottom panels and graph, $n = 3$) parental cell lines treated with scramble or CLN3-siRNA and immunostained for p73 protein. Nuclei are outlined by the yellow dashed line using Hoechst staining. Scale bar 10 μm. (I) Confocal fluorescence images of CLN3$^{KO}$ cells transfected with a control or two siRNAs against the YAP1 protein and immunostained for p73 protein ($n = 3$). Nuclei are outlined by the yellow dashed lines using Hoechst staining. Scale bar 10 μm. On the right, quantification of the mean intensity of p73 at the nucleus of at least 450 cells per each condition. (J) Representative immunoblot image and quantification of p73 co-immunoprecipitated with YAP1 protein in Wt and CLN3$^{KO}$ cells ($n = 5$). All the results are mean ± SEM. In the violin plots, orange dots represent the mean of each individual experiment. Statistical analysis with one-way ANOVA followed by Dunnett's multiple comparisons test (H, I), and Student's unpaired *t*-tests when comparing two experimental groups (A, B, D–J). Source data are available online for this figure.

silenced cells, in CLN3-silenced mouse neurons and in CLN3-KO brain.

Interestingly, the phosphorylation of YAP1$^{Y357}$ was also reported to increase the interaction between YAP1 and p73, which is important for YAP1 pro-apoptotic signaling (Levy et al, 2008). We therefore investigated whether p73 levels were increased in the nucleus of CLN3-KO cells. Our results show an increase in p73 nuclear localization in CLN3-KO cells (Fig. 2F). To further explore if the nuclear localizations of YAP1 and p73 were correlated, we co-stained cells with antibodies against YAP1 and p73. We observed that the percentage of cells with high nuclear YAP1 and p73 levels was just ~15% in Wt but >50% in CLN3-KOs (Fig. 2G, upper-right quadrant). Accordingly, after transient silencing of CLN3 in HEK293T and ARPE19 cells, we observed an increase in nuclear p73 levels in both cell lines (Fig. 2H), indicating that p73 nuclear localization is a specific response to loss of CLN3 in CLN3-KO cells, and not an indirect effect. Next, to determine if the accumulation of p73 in the nucleus was caused by YAP1, we transiently downregulated YAP1 in CLN3-KO cells, which led to a robust decrease in the nuclear localization of p73 (Fig. 2I). Additionally, by immunoprecipitation assays in whole cell extracts, we demonstrated that p73 interaction with YAP1increases in CLN3-KO cells when compared to Wt cells (Fig. 2J). Overall, these results show that the loss of CLN3 results in increased phosphorylation of YAP1 at Y357, which promotes its nuclear localization and recruitment of p73.

Finally, given that loss of CLN3 affects the functioning of lysosomes (Calcagni' et al, 2023; Nyame et al, 2024), we sought to clarify if the increase in pYAP1$^{Y357}$ and interaction with p73 may be a general consequence of lysosomal dysfunction or a specific event caused by loss of CLN3. To this end, we tested two human cell lines with prominent lysosomal defects, namely HeLa cells deficient in alpha-glucosidase (GAA-kd; enzyme essential for glycogen degradation in lysosomes) or lysosomal protease cathepsin B (CTSB-kd), which we previously characterized (Agostini et al, 2024). We showed that these lines have lower levels of pYAP1$^{Y357}$ compared to the respective scrambled control cells (Fig. EV3J). Furthermore, there was no enrichment of p73 in the nucleus of GAA-kd or CTSB-kd compared to the scrambled controls (Fig. EV3K). Since lysosomal defects can also result in perturbations of the autophagic

pathway, we tested if silencing Atg5, a protein involved in the early stages of autophagosome formation, would affect pYAP$^{Y357}$ signaling. In Atg5-silenced HeLa cells, we observed no changes in the levels of pYAP1$^{Y357}$ (Fig. EV3L). We further tested several primary fibroblasts from patients with different lysosomal storage diseases, including Gaucher disease (GBA mutation), Pompe disease (GAA mutation), Niemann-Pick type A (SMPD1 mutation) or type C (NPC1 mutation), and only in fibroblasts with NPC1 mutation, we found an increase in pYAP1$^{Y357}$ (Fig. EV3M). Finally, to test if loss of lysosomal acidification, a broader perturbation of lysosomal biology, would impact YAP1$^{Y357}$ phosphorylation, we treated WT and CLN3-KO cells with Bafilomycin (BafA) and observed that there was no increase in WT or CLN3-KO cells (Fig. EV3N). Altogether, these results suggest that the activation of YAP1 signaling is a consequence triggered by the loss of CLN3 and affected by impairment of lysosomal cholesterol homeostasis, but it is not a generic consequence of impaired lysosomal function, or of decreased autophagy.

## YAP1 activation in CLN3-KO cells depends on c-Abl activity

Given the importance of Y357 phosphorylation for the role of YAP1 in CLN3-KO cells, we proceeded to determine the underlying mechanism. Considering that phosphorylation of YAP1 residue Y357 is mediated by the c-Abl tyrosine kinase (Levy et al, 2008), we started by assessing total c-Abl levels in whole cell extracts of Wt and CLN3-KO cells, and observed an increase in CLN3-KO HEK293T cells (Fig. 3A). c-Abl continuously shuttles between the nucleus and the cytoplasm (Taagepera et al, 1998). We next determined if its nuclear localization is affected in CLN3-KO cells. Interestingly, we detected a robust increase in nuclear c-Abl in HEK293T and ARPE19 CLN3-KO cells when compared to the Wt cells (Figs. 3B and EV4A,B). To confirm that c-Abl was responding specifically to the absence of CLN3, we expressed CLN3-GFP in the HEK293T CLN3-KO cells, and observed a robust decrease in c-Abl nuclear localization (Fig. EV4C). We then sought to verify if c-Abl was also upregulated in the absence of CLN3 in vivo, and observed

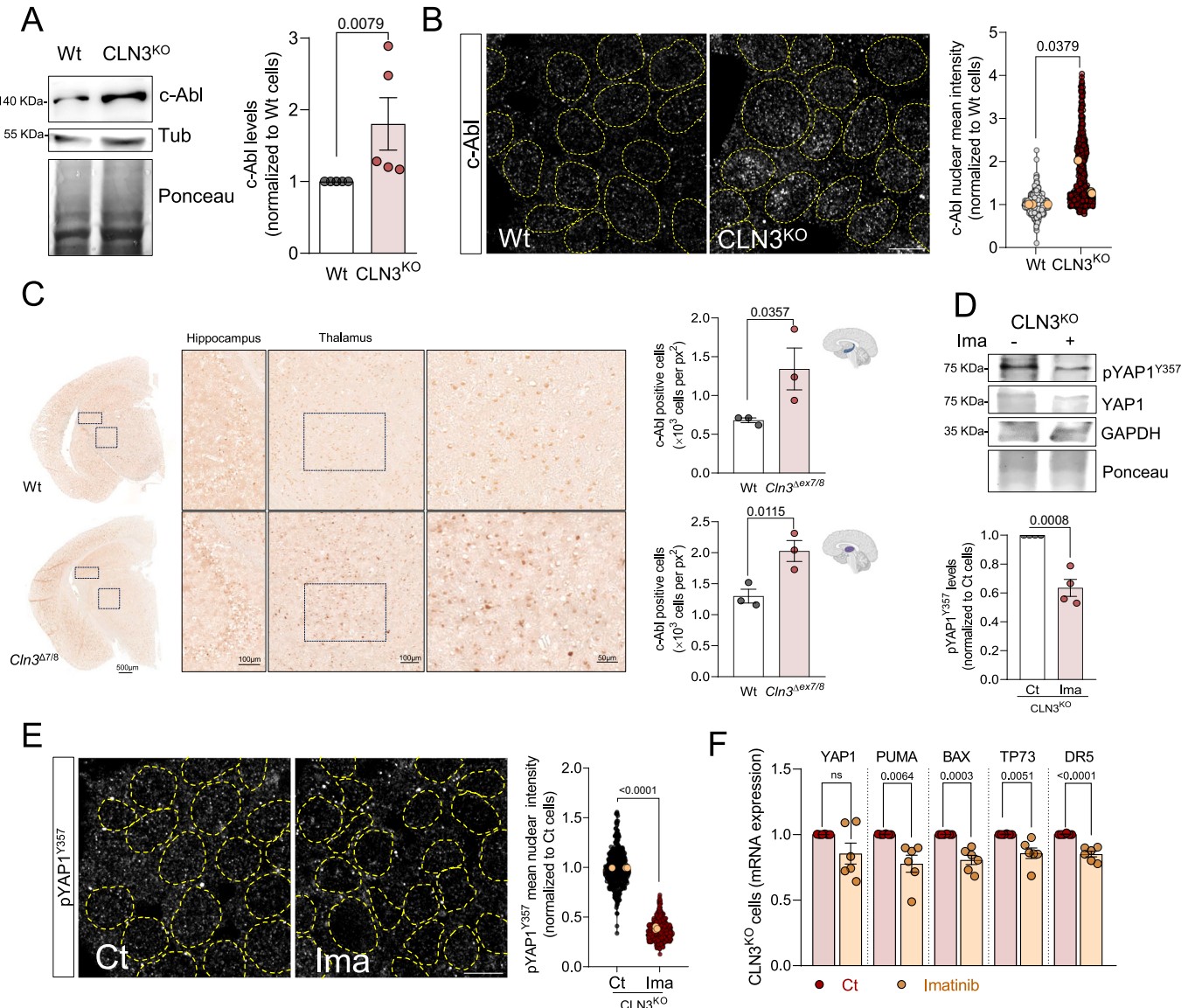

**Figure 3. Loss of CLN3 triggers activation of c-Abl.**

(A) Representative immunoblot image and quantification of c-Abl protein levels in Wt and CLN3^KO cells. Tubulin and Ponceau were used as loading controls ($n = 3$). (B) Confocal fluorescence images of HEK293T WT and CLN3^KO cells immunostained for c-Abl protein. Nuclei are outlined by the yellow dashed line using Hoechst staining. Scale bar 10 μm. On the right, quantification of the mean intensity of c-Abl at the nucleus from at least 100 cells per condition in each independent experiment, with a total of at least 500 cells ($n = 3$). The orange dots represent the mean of each independent experiment. (C) Immunohistochemistry analysis of c-Abl staining of control (Wt) and $Cln3^{\Delta7/8}$ animals; right panels are cropped images from thalamus; quantification of the number of positive nucleus to c-Abl in the total area from the hippocampus (upper panel) and the thalamus (bottom panel) ($n = 3$). (D) Representative immunoblot image and quantification of pYAP1^Y357 in CLN3^KO cells, control or treated for 24 h with 0.1 μM imatinib (c-Abl inhibitor). GAPDH and Ponceau were used as loading controls. (E) Confocal fluorescence images of HEK293T CLN3^KO control or treated with imatinib (24 h, 0.1 μM) and immunostained for pYAP1^Y357 protein. Nuclei are outlined by the yellow dashed line using Hoechst staining. Scale bar 10 μm. On the right, quantification of the mean intensity of c-Abl at the nucleus from at least 100 cells per condition in each independent experiment, with a total of at least 350 cells. The orange dots represent the mean of each independent experiment. (F) mRNA levels of YAP1 target genes involved in apoptosis in HEK293T CLN3^KO cells untreated or treated for 24 h with imatinib ($n = 6$). All the results are mean ± SEM. Statistical differences between the two conditions were assessed using an unpaired $t$-test. Source data are available online for this figure.

a prominent increase in c-Abl staining in both the hippocampus and thalamus of $Cln3^{\Delta7/8}$ mice (Fig. 3C).

Further, we tested if c-Abl was driving YAP1 hyperactivity by treating CLN3-KO cells with the c-Abl inhibitor imatinib (0.1 μM, 24 h). Our data revealed decreased levels of pYAP1^Y357 in whole cell

extracts of CLN3-KO HEK293T cells incubated with imatinib (Fig. 3D), with no significant changes in YAP1 total levels. Importantly, inhibition of c-Abl also resulted in lower amounts of pYAP1^Y357 localized to the nucleus (Fig. 3E). Accordingly, the transcript levels of YAP1 pro-apoptotic targets also decreased in

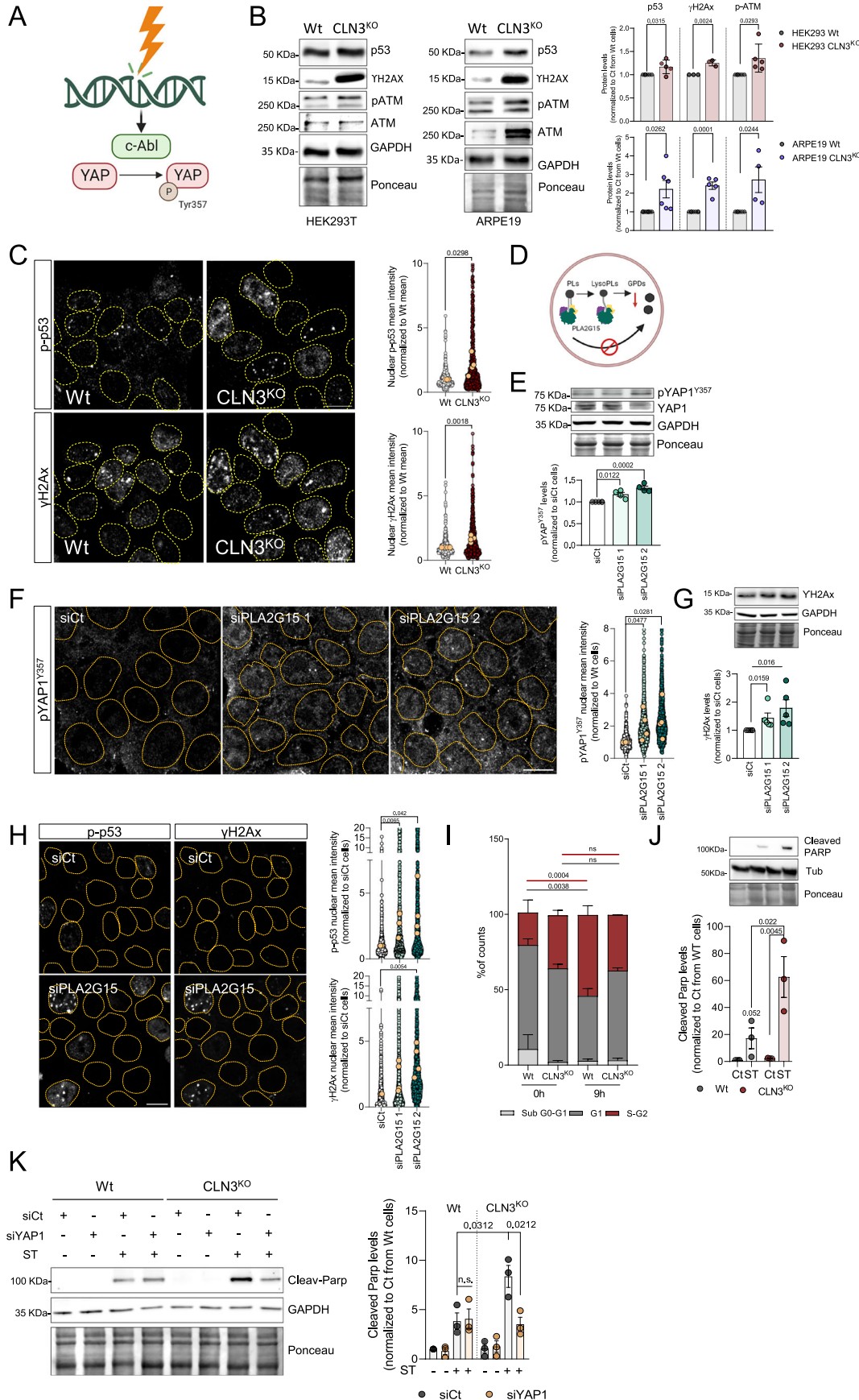

**Figure 4. Accumulation of DNA damage in CLN3-KO cells drives c-Abl and nuclear dysmorphism.**

(A) Schematic representation of c-Abl activation in response to DNA damage. C-Abl directly phosphorylates YAP1 at the position Y357 in response to DNA damage. (B) Representative immunoblot image and quantification of p53, γH2AX, p-ATM and ATM protein levels in HEK293T ($n = 3$–5) and ARPE19 ($n = 4$–6) Wt and CLN3[KO] cells. GAPDH and ponceau were used as loading controls. (C) Confocal microscopy images of HEK293T Wt and CLN3[KO] cells immunostained for p-p53 and γH2AX proteins. Nuclei are outlined by the yellow dashed line using Hoechst staining. Graphs are the quantification of the mean intensity of p-p53 (upper panel, $n = 4$) and γH2AX (bottom panel, $n = 4$) at the nucleus from at least 100 cells per condition in each independent experiment, with a total of at least 450 cells. Scale bar 10 μm. (D) Schematic representation of glycerophosphodiesters (GPDs) synthesis inhibition by affecting the activity of lysosomal lipases that mediate the hydrolysis of phospholipids (PLs) into GPDs. (E) Representative immunoblot image and quantification of pYAP1[Y357] protein levels in Wt cells transfected with control (siCt) and two different siRNAs against lysosomal lipase A2 (PLA2G15) ($n = 4$). Ponceau was used as a loading control. (F) Confocal fluorescence images and quantification of HEK293T Wt and cells depleted from PLA2G15 of pYAP1[Y357] nuclear levels ($n = 4$). Nuclei are outlined by the yellow dashed line using Hoechst staining. Scale bar 10 μm. (G) Representative immunoblot image (above) and quantification (below) of γH2AX protein levels in control cells and cells silenced for PLA2G15 ($n = 5$). Ponceau was used as loading controls. (H) Confocal fluorescence images and quantification (upper graph for p-p53 ($n = 4$), bottom for γH2AX ($n = 4$)) of HEK293T parental cells transfected with siCT or siRNA against PLA2G15. Nuclei are outlined by the yellow dashed line using Hoechst staining. Scale bar 10 μm. (I) Cell cycle quantification of Wt and CLN3[KO] cells stained with propidium iodide (PI) after cell synchronization with an overnight starvation. The percentage of cells in each cycle phase was quantified by flow cytometry ($n = 3$). (J) Representative immunoblot of cleaved PARP and quantification of HEK293T (Wt and CLN3[KO]) cells with or without staurosporine (ST) treatment (4 h) ($n = 3$). (K) Representative immunoblot image and quantification of cleaved PARP protein levels in HEK293T (Wt and CLN3[KO]) cells transfected with siCt or siYAP and treated or untreated with staurosporine (ST) treatment (4 h) ($n = 3$). GAPDH and ponceau were used as loading controls. (L) Schematic representation of our working model. All the results are mean ± SEM. In the violin plots, the orange dots indicate the mean value of each independent experiment. Statistical analysis with one-way ANOVA followed by Dunnett's multiple comparisons test (E, F, H), Mann–Whitney test (G), Tukey's multiple comparisons test (I, J), and Student's unpaired *t*-tests when comparing two experimental groups (B, C). In (K), an unpaired *t*-test was used for direct comparison between the specific conditions. Source data are available online for this figure.

imatinib-treated CLN3-KO cells (Fig. 3F). These results show that YAP1 function and its pro-apoptotic signaling in CLN3-KO cells are dependent on c-Abl activity.

## Activation of c-Abl in CLN3-KO cells in response to DNA damage

Considering that DNA damage is one of the major stimuli leading to c-Abl activation (Shaul and Ben-Yehoyada, 2005) and given that our transcriptome data indicated that DNA damage/repair pathways are affected in CLN3-KO cells (Fig. 1B), we next investigated the presence of DNA stress response in the absence of CLN3 (Fig. 4A). We first evaluated γH2Ax, a marker of unresolved DNA double-strand breaks, as well as p53 and phospho-ATM[Ser1981], both involved in DNA repair signaling, and observed a robust increase in the levels of these three proteins in both HEK293T and ARPE19 CLN3-KO cells (Fig. 4B). We also observed an increase in the fluorescence intensity of p53[Ser15] and γH2Ax in the nucleus of both CLN3-KO cell lines (Figs. 4C and EV5A,B). Moreover, by expressing CLN3-GFP in CLN3-KO cells, the accumulation of double-strand breaks was reverted, assessed by quantification of γH2Ax (Fig. EV5C). Furthermore, when CLN3 was transiently silenced in ARPE19 and HEK293T cells, we also observed an increase in γH2Ax (Fig. EV5D,E). This result confers further validity to the pathway analysis of the CLN3-KO transcriptome, which identified ATM signaling (Fig. 1B), a key pathway of DNA damage response, as one of the most enriched pathways in the CLN3-KO transcriptome.

Considering the role of CLN3 on the lysosomal efflux of glycerophosphodiesters (GPDs) (Laqtom et al, 2022), we next investigated whether an impairment of GPDs efflux from the lysosomes was causative of the DNA damage observed in CLN3-KO cells. To address this question, we silenced the lysosomal phospholipase A (PLA2G15) (Fig. EV5F), which is the first enzyme processing glycerophospholipids in the lysosome (Fig. 4D), expecting that in the absence of PLA2G15, no GPDs would be generated in the lysosomal lumen. By disrupting the pathway upstream of CLN3, we can test if the decreased efflux of GPDs from the

lysosomes is sufficient to trigger DNA damage and, consequently, activate YAP1 pro-apoptotic signaling. Interestingly, PLA2G15 silencing in HEK293T cells was enough to increase the total levels of pYAP1[Y357] (Fig. 4E) and in its nuclear localization (Fig. 4F), as well as to increase nuclear γH2Ax (Fig. 4G,H). Of note, phosphorylated p53[Ser15] was also enriched in the nucleus of PLA2G15-silenced HEK293T cells (Fig. 4H).

Finally, we investigated the impact of CLN3 loss on cell cycle and apoptosis. Our data demonstrated that, after cell synchronization by serum starvation, CLN3-KO cells presented cell cycle arrest, with increased retention of the cells in G1 phase and a decrease in the percentage of cells in S phase after 9 h of restoring the serum levels (Fig. 4I). In addition, to determine this cell cycle arrest and the observed increased in pro-apoptotic YAP1 transcriptional targets translate into increased apoptosis susceptibility, we assessed the levels of cleaved PARP, a substrate of caspase-3 and a well-known marker of apoptosis (Chaitanya et al, 2010), in CLN3-KO and Wt cells. Under basal conditions, we observed similarly low levels of cleaved PARP in Wt and CLN3-KO cells (Fig. 4J), indicating low apoptotic activity. However, when the cells were subjected to a mild treatment with the apoptosis inducer staurosporine (2 μM, 4 h), the levels of cleaved PARP significantly increased in CLN3-KO cells compared to Wt (Fig. 4J). Importantly, when cells were silenced for YAP1 and treated with staurosporine, Wt cells presented similar levels of cleaved PARP, whereas silencing YAP1 in CLN3-KO showed a significant reduction of cleaved PARP compared with si-scrambled transfected cells (Figs. 4K and EV5G). This protective effect of blocking YAP1-mediated pro-apoptotic signaling in CLN3-KO cells was also verified by c-Abl inhibition: treating CLN3-KO with imatinib was enough to diminish the cleaved PARP levels upon staurosporine treatment, with no observed effect on Wt cells (Fig. EV5H). These results suggest that, while CLN3-KO cells in basal conditions are still protected from apoptosis, these are more susceptible to cell death when subjected to mild stress, likely due to YAP1 pro-apoptotic signaling.

Despite the recent finding that CLN3 protein is involved in the pathway that exports glycerophosphodiesters (GPDs) from the lysosome (Laqtom et al, 2022), the consequences of lysosomal GPD

export impairment on cellular lipid homeostasis are still unclear. Our results suggest that this process may affect the nuclear lipid homeostasis, impacting the mechanisms involved in DNA damage response. Once exported from the lysosome, GPDs are further metabolized by glycerophosphodiesterases (GDEs). There are seven GDEs in mammals (Corda et al, 2014), with different intracellular localizations, several of which are associated with cellular membranes in proximity to the nucleus. The nucleus has the enzymatic machinery to synthesize phospholipids (Fujimoto, 2024), therefore the GPDs could conceivably be used as substrates in this pathway. The observed increase in DNA damage in response to CLN3 or PLA2G15 depletion may indicate that metabolic products from lysosomal lipid digestion are important for lipid homeostasis and possible nuclear membrane remodeling necessary for DNA homeostasis or clearance of damaged DNA. Notably, PLA2G15 was recently implicated in the catabolism of BMP (Abe et al, 2024; Nyame et al, 2024), an important phospholipid necessary for the digestion of lipids in the lysosomal lumen. It is therefore possible that the effect of PLA2G15 on DNA damage and downstream c-Abl/YAP1 signaling is not exclusively due to the perturbation of GPD metabolism and its efflux from the lysosomes.

The GPDs released from the lysosomes may also be transferred to the endoplasmic reticulum (ER), via lysosome-ER contact sites, and then laterally diffuse towards the nuclear envelope, which is continuous with the ER. The involvement of lysosome-ER contact sites in the regulation of the lipid composition of the nuclear envelope and of DNA maintenance is supported by the presence of DNA damage in the subventricular zone of NPC1-KO mice (Seo et al, 2012). NPC1 is involved in lysosomal cholesterol and sphingomyelin efflux (Newton et al, 2018; Wanikawa et al, 2020), and its loss-of-function results in decreased membrane contact sites between the lysosomes and the ER (Höglinger et al, 2019). Decreased association between these two organelles was already shown to impact the lipid composition of ER membranes (Agostini et al, 2024). Given that the nuclear envelope is a subdomain of the ER membrane, changes in ER lipid composition may lead to alterations in the lipid profile of the nuclear envelope. Interestingly, under conditions of intracellular cholesterol accumulation due to NPC1 loss-of-function, c-Abl inhibition has been shown to protect cells and promote cellular clearance (Contreras et al, 2020). Our results demonstrate a role for lysosomal lipid metabolism in the regulation of c-Abl activity. Further research is needed to understand the molecular basis by which lysosomal lipid metabolism impacts cellular signaling through modulation of ER and/or nuclear membrane dynamics.

# Conclusion

We show here that loss of CLN3, the lysosomal protein affected in Batten disease, leads to DNA damage accumulation and pro-apoptotic signaling in human cells and in the mouse brain.

The detection of DNA damage markers downstream of impairment of lysosomal phospholipid catabolism opens the possibility for novel disease-modifying therapeutic strategies for patients with Batten disease caused by mutations in CLN3. Our data show that one of the proximal consequences of DNA damage is the activation of the kinase c-Abl, and that inhibition of c-Abl ablates YAP1 pro-apoptotic signaling. It remains to be tested in vivo, in models of CLN3 loss-of-function, whether inhibition of c-Abl would delay the onset of symptoms or their

progression. Strikingly, there are several commercially available drugs that target c-Abl, which, if effective in CLN3 loss-of-function, might constitute a therapeutic approach for Batten disease, which is currently incurable.

Together, our findings highlight the activation of YAP1 pro-apoptotic signaling via c-Abl activation as a critical pathway in lysosomal dysfunction resulting from impaired GPD efflux, and identify a novel target for possible therapies in Batten disease.

# Methods

**Reagents and tools table**

| Reagent/resource | Reference or source | Identifier or catalog number |
|---|---|---|
| **Experimental models** | | |
| Human embryonic kidney (HEK) | ATCC | CRL-11268 |
| ARPE19 CLN3$^{KO}$ | Kindly given by Alessia Calcagni' (Calcagni' et al, 2023) | - |
| ARPE19 Wt | ATCC | CRL-2302 |
| Cln3$^{Δ7/8}$ knock-in mice | (Cotman, S. L et al, Human molecular genetics, 2002) | - |
| C57BL/6J mice | | |
| **Recombinant DNA** | | |
| CLN3-GFP | Addgene | #78110 |
| pFUGW-GFP | A kind gift from Dr. Luis Ribeiro | |
| **Antibodies** | | |
| Anti-YAP | Cell Signaling Technology | Cat#4912S |
| Anti-pTyr357-YAP | Sigma-Aldrich | Cat# Y4645 |
| Anti-p73 | Cell Signaling Technology | Cat#14620S |
| Anti-c-Abl | Cell Signaling Technology | Cat#2662P |
| Anti-H2Ax | Cell Signaling Technology | Cat#9718S |
| Anti-p53 | Proteintech | Cat#10442-1-AP |
| Anti-p-p53 | Cell Signaling Technology | Cat# 9286P |
| Anti-mouse monoclonal anti-Tubulin | Sigma-Aldrich | Cat# T6199 |
| Anti-rabbit monoclonal anti-GAPDH | Cell Signaling Technology | Cat#5174 T |
| Mouse polyclonal anti-hnRNPA2B1 [B-7] | Santa Cruz | Cat#sc-374053 |
| Anti-cleaved-PARP | Cell Signaling Technology | Cat#5625 P |
| Anti-ATM | Cell Signaling Technology | Cat#2873 |
| Anti-p-ATM (Ser1981) (D6H2) | Cell Signaling Technology | Cat#5883 |
| Anti-lamin A/C | Santa Cruz | Cat#376248 |
| HRP-conjugated goat anti-mouse IgG | Cell Signaling Technology | Cat#7076S |
| HRP-conjugated goat anti-rabbit IgG | Cell Signaling Technology | Cat#7074P2 |
| Alexa 647-conjugated goat anti-rabbit IgG | Invitrogen | Cat#A21245 |
| Alexa 647-conjugated goat anti-rabbit IgG | Invitrogen | Cat#A21245 |
| Alexa 647-conjugated goat anti-rabbit IgG | Invitrogen | Cat#A21245 |
| Alexa 568-conjugated goat anti-mouse IgG | Invitrogen | Cat#A11031 |
| **Oligonucleotides and other sequence-based reagents** | | |
| h_siCLN3_1 | UUGUUCUUUCAAGGUCU AUUCUUUAUAGACCUUGAA | |

| Reagent/resource | Reference or source | Identifier or catalog number |
|---|---|---|
| h_siCLN3_2 | GCAGUACCGAUGGUACC AUAGCAUCUGGUACCAUCG | |
| h_siYAP_1 | ACGGUAGAUAUUACUGAC AAUUCAUCAGAUAAUAU | |
| h_siYAP_2 | GCUGCCACCAAGCUAGAU UUUCUUUAUCUAGCUUGG | |
| h_siPLA2G15_1 | GUAUCUGGAUUCUGGCAA ACUUUUAUUGCCAGAAUC | |
| h_siPLA2G15_2 | AGACCGAAAGCUACUUCACA GAUUGUGAAGUAGCUU | |
| h_PUMA | F- ATCAATCCCATTGCA TAGGTTTAG | R-ACTAAGGCTG GGGCGCTTC |
| h_TP53AIP1 | F-GCTCAGACACACACACCT | R-GGCCTGTCTCT AAGCACTGT |
| h_Bax | F-TGTTTTTCTGACGGCAACTTC | R-ATCAGTTCCG GCAACCTTG |
| h_TP73 | F-CCCACCACTTTGAGGTCACT | R-GCGATCTG GCAGTAGAGTT |
| h_DR5 | F-TGATTCAGGTGAAGTGGAGC | R-CGACCTTGAC CATCCCTCTG |
| h_YAP1 | F-TAGCCCTGCGTAGCCAGTTA | R-CATGCTTAGT CCACTGTCTGT |
| h_CLN3 | F- CGCCCACGACATCCTTAGC | R-AGCAGCCGTA GAGACAGAGTT |
| **Chemicals, enzymes and other reagents** | | |
| SYBR green master mix | BioLabs | M3003E |
| ECL prime western blotting detection reagent | Cytiva, Amersham | RPN2236 |
| Pierce BCA protein assay kit | Thermo Fisher Scientific | Cat #23225 |
| Nuclear/cytosol fractionation kit | Abcam | AB289882 |
| NucleoBond isolation kit | Macherey-Nagel | 740410.100 |
| iScript cDNA synthesis kit | Bio-Rad | 170-8891 |
| Staurosporine | MedChemExpress | HY-15141 |
| Imatinib | Sigma | CDS022173 |
| Lipofectamine 3000 | Ivitrogen | L3000-015 |
| PFA | Thermo Fisher | P-0840-53 |
| PBS | Sigma | P4417-100TAB |
| Saponine | Serva | 34655.01 |
| BSA | Fisher Scientific | BP9704-100 |
| Hoechst | | |
| Mowiol | Sigma | SML1027- 10MG |
| HEPES | Thermo Fisher | 81381-250 G |
| SYBR green master mix | BioLabs | M3003E |
| Halt TM protease & Phosphatase single-use Inhibitor Cocktail (100X) | Thermo Scientific | 78440 |
| Protein G Sepharose beads | Cytiva | 17061801 |
| Tween 20 | Thermo Fisher | BP337-500 |
| Tris base | Enzymatic | BP152-5 |
| NaCl | Roth | Nr.P029.3 |
| Sodium dodecyl sulfate ultrapure | Sigma | L4509-500G |
| Ponceau staining | AppliChem | A2935,0500 |
| 2-Mercaptoethanol | Sigma-Aldrich | Cat#M6250-100ml |
| Bafilomycin A | Invivogen | Cat#tlrl-baf1 |
| Bis-Acrylamide | Thermo Fisher Scientific | Cat#221897 |
| Bovine serum Albumin | Sigma-Aldrich | Cat#A8806-5G |
| DMEM | Sigma-Aldrich | Cat# D6546-6X500ML |
| DMEM-F12 | Gibco | Cat# 21331-020 |
| DMSO | Sigma-Aldrich | Cat#D26650-100ML |
| Ethanol | Sigma-Aldrich | Cat#3221.2.5L-M |
| FBS | Sigma-Aldrich | Cat# F7524-100ML |
| Gelatin | Sigma-Aldrich | Cat#G7765 |
| Methanol | Sigma-Aldrich | Cat#32213.2.5L-M |
| PageRuler Prestained Plus | Thermo Fisher Scientific | Cat#26619 |
| Penicillin-streptomycin | Sigma-Aldrich | Cat# P0781-100ML |
| Poly-L-lysin | Sigma-Aldrich | Cat#A38904-01 |
| Skim milk powder | Sigma-Aldrich | Cat#70166-500 G |
| Sodium dodecyl sulfate | Thermo Fisher Scientific | Cat#15825-017 |
| Sodium hydroxide | Sigma-Aldrich | Cat#S5881-1KG |
| 2.5% glutaraldehyde | Sigma-Aldrich | G6403 |
| 0.1 M sodium cacodylate buffer (pH 7.2) | Agar Scientific | |
| 1 mM calcium chloride | Alfa Aesar | L13191 |
| 1% osmium tetroxide | Sigma | |
| 2% molten agar | AGAG-00P-500 | Labkem |
| Epoxy embedding kit | Fluka Analytical | |
| Lead citrate 0.2% | Sigma | S4641-25G |
| ECL prime western blotting detection reagent | Cytiva | Cat#RPN2236 |
| TEMED | Carl Roth | Cat#2367.1 |
| Trypsin-EDTA | Sigma-Aldrich | Cat# T3924-100ML |
| OCT | Tissue-Tek | 4583 |
| BLOXALL | Vectorlab | SP-6000 |
| MACH4 HRP-Polymer | Biocare Medical | M4U534 G |
| NovaRED | Vector Labs | SK-4800 |
| Aqueous mounting medium | Biorad | BUF058B |
| Clasto-Lactocystin ß-Lactone | Merck | 426102 |
| Xmu-mp-1 | TargetMoi | TM-T4212 |
| HBSS with CaCl2 and MgCl2 | Gibco | 24020-117 |
| L-Cystein | Thermo Scientific | J64745.22 |
| Neurobasal media | Gibco | 2E + 07 |
| Minimum essential medium Eagle (MEM) | Sigma-Aldrich | M0268 |
| Horse Serum | Gibco | 2E + 07 |
| Poly-D-Lysine | Sigma-Aldrich | P7886 |
| Glutamine | Gibco | 3E + 07 |
| Gentamicin | Gibco | 2E + 07 |
| NeuroCult(TM) SM1 Neuronal Supplement | StemCell | 5711 |
| **Software** | | |
| Fiji Image J1 software | | version 1.53 q |
| Partek Software Suite | | |
| CFX Opus 384 Real-Time PCR System with One software | Applied Biosystems | v2.2.2 |
| Prism | GraphPad (Prism 10) | https://www.graphpad.com/ |
| Excel | Microsoft | https://products.office.com/en-us/explore-office-for-home |
| Imaris | 9.9.0 Oxford Instruments, Zurich, Switzerland | |
| **Other** | | |
| Nuclear/cytosol fractionation kit | Abcam | AB289882 |

| Reagent/resource | Reference or source | Identifier or catalog number |
|---|---|---|
| FITC | Merck | 60842-46-8 |
| iScript cDNA synthesis kit | Bio-Rad | 170-8891 |
| Alexa Fluor 647 | Thermo Fisher | D22914 |
| Magic-red | Bio-Rad | ICT937 |
| GasPaKTM EZ – Anaerobe Gas Generating pouch system | BD | 260683 |
| Zeiss LSM 710 | Carl Zeiss AG | |
| BioRender | | |

## Cell culture

Human embryonic kidney HEK293T cells were cultured in Dulbecco's modified Eagle's medium (DMEM) supplemented with 10% FBS and Penicillin/Streptomycin (100 U/ml:100 µg/ml), at 37 °C under 5% CO2. ARPE19 were cultured as previously described (Calcagni' et al, 2023).

## Antibodies

The following antibodies were used: anti-YAP1 (IB 1:1000); anti-pTyr357-YAP (IB 1:1000, IF 1:100); anti-p73 (IB 1:1000, IF 1:100); anti-c-Abl (IB 1:1000, IF 1:100); anti-H2Ax (IB 1:1000, IF 1:100); anti-p53 (IB 1:1000, IF 1:100); anti-p-p53 (1:1000, IF 1:100); anti-mouse monoclonal anti-Tubulin (IB 1:1000); anti-rabbit monoclonal anti-GAPDH (IB 1:1000); mouse polyclonal anti-hnRNPA2B1 [B-7] (IB:1000); anti-cleaved-PARP (IB 1:1000); anti-ATM (B 1:1000); anti-p-ATM (Ser1981) (D6H2) (IB 1:1000); anti-lamin A/C (IF 1:100); HRP-conjugated goat anti-mouse IgG (IB 1:5000); HRP-conjugated goat anti-rabbit IgG (IB 1:5000); Alexa 647-conjugated goat anti-rabbit IgG (IF 1:500); Alexa 568-conjugated goat anti-mouse IgG (IF 1:500); mouse monoclonal anti-LAMP1 [H4A3] (IB 1:1000, IF 1:100); anti-CLN3 (kindly given by Alessia Calcagni' (Calcagni' et al, 2023); anti-LC3 (IB 1:1000, IF 1:100); anti-PLA2G15 (IB 1:1000).

## DNA constructs

The CLN3-GFP plasmid was obtained from Addgene (#78110), a kind gift from Dr. Thomas Braulke.

## Generation of CLN3-KO HEK cells

To knock out CLN3 in HEK293T cells, we used a CRISPR–Cas9 approach. A guide RNA (gRNA) corresponding to the first exon of CLN3 protein was designed in CLC Workbench 7 based on its genomic DNA sequence available on the Ensembl (Forward 5'-CACCTGCGATGGGAGGCTGTGCAGGCTCG-3' and Reverse 5'-AAACCGAGCCTGCACAGCCTCCCATCGCA-3'; IDT Technologies). Plasmid pSpCas9(BB)-2A-Puro (PX459) was digested with BbsI. The band at 9 kb, corresponding to the total size of the plasmid (9175 bp), was cut out from the gel and purified utilizing NucleoSpin Gel and PCR Clean-up kit, and subsequently ligated with the gRNA, followed by transformation of E. coli strain XL10-Gold ultracompetent cells, from which the plasmid was

subsequently purified. To generate CLN3-KO in HEK293T cells, transfection of parental HEK293T cells was performed using Lipofectamine 3000 according to the supplier's instructions. After 48 h the media in transfected and natural puromycin resistance control plates were changed to a growth medium with 2 µg/ml of puromycin added to start antibiotic selection. Selection media were changed every 2–3 days until all Wt cells in the control plate were dead as determined by observation under the microscope. Single-cell colonies were obtained by flow cytometry at the Core Facility Cell-Sorting of the University Medical Center Göttingen, and DNA was sequenced to determine colonies with CLN3-KO.

## Mice

Cln3$^{\Delta7/8}$ knock-in mice generation was previously described (Cotman et al, 2002). Mice were genotyped using the following primers: 5'-TTTGTTCTGCTGGGAGCTTT-3', 5'-CAGTCTCTGCCTCGT TTTCC3'-, 5'-GACAAGAGCACTGAGGAAGAT-3', 5'-AGGA AGGAATGGGGAGACTGA-3', resulting in a 600 bp band in the case of Cln3$^{\Delta7/8}$ mice, and a 450 bp band for the wild type allele. Mice used for experiments were maintained in a C57BL/6J strain background.

All experiments were approved by the Committee on Animal Care at Baylor College of Medicine (protocol AN-5280) and conform to the legal mandates and federal guidelines for the care and maintenance of laboratory animals.

## Cell treatments

After 24 h of seeding, cells were treated with: 2 µM staurosporine for 4 h; 0.1 µM imatinib for 24 h; and liposomes 1.5 mM of total lipids for 24 h.

## Cell transfection and siRNA-mediated knockdown

DNA and siRNA transfections were performed using Lipofectamine 3000 according to the manufacturer's instructions. siRNA target sequences were: CLN3,, YAP1 and PLA2G15. After 24 h of cell seeding, 20 nmol/L siRNA was complexed with the transfection reagent and incubated at 37 °C. After an overnight incubation, the transfection media was replaced with fresh media and the experiments were performed after 72 h. Non-targeting control sequences were used as controls.

## Immunofluorescence and cell imaging

Cell lines were grown on gelatine-coated coverslips were fixed in 4% (w/v) PFA for 10 min, followed by 20 min permeabilization and blocking with PBS with 0.05% (w/v) saponin and 1% (w/v) BSA (blocking solution). Coverslips were then incubated overnight with primary antibodies prepared in the blocking solution at 4 °C. Next, coverslips were washed with PBS and incubated with Alexa Fluor-conjugated secondary antibodies for 1 h, at room temperature. Nuclei were stained with Hoechst. Finally, coverslips were mounted with Mowiol 4-88. In order to confirm their specificity, all the antibodies were tested through immunoblot analysis and secondary antibody controls, where primary antibodies were not added. Images were acquired on a confocal microscope (Zeiss LSM 710; Carl Zeiss AG) by using a Plan-Apochromat 63x/1.4 oil DIC M27

objective. For each experimental condition, at least 5 different fields were imaged. The imaging of an immunostained cell was performed using the same settings.

For live imaging, cells were seeded in an Ibidi chamber. After an overnight transfection, lipid and Dextran Alexa Fluor 647 loading, cells were washed out for 4 h with complete DMEM without phenol red supplemented with 10 mM HEPES. Live cells were imaged using the LSM 710 confocal system and a 63x/1.4NA objective. A single stack was acquired by time-lapse imaging, through the acquisition of one z-stack image per 30 s for a total of 10 min.

## Immunohistochemistry

After embedding brains in OCT, blocks were sectioned coronally at 10 μm. For immunohistochemistry procedures, sections were washed with 1x Tris-buffered saline (TBS 1X), blocked with 5% goat serum, 0.25% Triton in TBS for 60 min, then treated for 10 min with the Bloxall solution to quench endogenous peroxidases. Next, sections were washed in TBS and incubated overnight at 4 °C with primary antibodies diluted in the blocking solution (5% goat serum and 0.25% Triton in TBS). The next day, sections were washed in TBS, and incubated with the MACH4 HRP-Polymer for 1H. Slides were rinsed in TBS 1X, and immune reactions were revealed using the Peroxidase (HRP) Substrate, NovaRED. Sections were washed and mounted with permanent aqueous mounting medium. Anti-c-Abl (1:200), Anti pYAP1(1:100)

## Image analysis

Fluorescent image analysis was performed on the original data using Fiji Image J1 software (version 1.53 q). For quantification of nuclear levels of YAP1, pTyr357-YAP1, p73, c-Abl, p-p53, nuclear regions were segmented by applying a threshold to Hoechst staining and using the "analyze particles" plug-in from Fiji (definition of our region of interest, ROIs). The background image was measured and removed from each nucleus.

## RNAseq

RNA sequencing was performed as described (Murdoch et al, 2016). RNAseq data analysis was performed using Partek Software Suite. RNAseq data were aligned to the reference genome mm10 by the BoWtie algorithm, and the transcripts were quantified using the human GRCh37 Ensembl assembly as reference. For differential expression analysis, Bonferroni multi-test correction was applied, and adjusted $p$ value <0.05 was considered significant. Ingenuity Pathway Analysis was used for assessment of transcription factor activity, as described (Yambire et al, 2019; Agostini et al, 2024; Murdoch et al, 2016). The data were deposited at Gene Expression Omnibus (GSE265831).

## Nuclear extracts

Nuclear extracts were performed using the Nuclear/cytosol fractionation kit following the manufacturer's instructions.

## Quantitative RT-PCR

RNA was extracted with the NucleoBond kit isolation kit. Reverse transcription was performed using the iScript cDNA synthesis kit.

Quantitative PCR was performed in a 384-well plate using the SYBR green master mix using the CFX Opus 384 Real-Time PCR System with CFX Maestro software. Hprt and ActA were used as housekeeping genes to normalize the expression. Target gene expression was determined by relative quantification (ΔΔCt method) to the housekeeping reference gene and the control sample. The primer sequences are indicated in Table EV1.

## Immunoprecipitation

Cell lysates were prepared in RIPA buffer, supplemented with protease and phosphatase inhibitors. Immunoprecipitations were performed using 500 μg of total protein lysates. The YAP1 primary antibody (0.5 μg) was incubated with the cells at 4 °C under rotation overnight. Protein G Sepharose beads were then added and incubated for 1 h, at 4 °C. After washing, complexes were eluted in Laemmli buffer, boiled for 5 min at 95 °C and analyzed by Western blot.

## Western blot

Cell extracts (50 μg of protein) were prepared using Laemmli buffer. Before the gel loading, samples were sonicated and boiled for 5 min at 95 °C. Subsequently, samples underwent separation via sodium dodecyl sulfate polyacrylamide gel electrophoresis (SDS-PAGE) and were transferred onto nitrocellulose membranes. Ponceau staining was conducted for total protein evaluation, followed by washing. Membranes were washed and then blocked with 5% (w/v) non-fat milk in Tris-buffered saline-Tween 20 (TBST; 20 mM Tris, 150 mM NaCl, 0.2% (v/v) Tween 20, pH 7.6) for at least 30 min. Primary antibodies were incubated overnight at 4 °C, followed by incubation with HRP-conjugated secondary antibodies for 1 h at room temperature. Protein detection was accomplished via chemiluminescence (ECL Prime Western Blotting Detection Reagent, Cytiva, Amersham, RPN2236) using an ImageQuant 800 from Cytiva. Quantification was performed on unsaturated images using Image Lab software.

## Lysosomal pH and degradative activity

For lysosome pH, cells were loaded with dextran conjugated with FITC (250 μg/mL) and dextran conjugated with Alexa Fluor 647 (50 μg/mL) overnight, followed by a 4 h chase before treatment with 100 nM bafilomycin A1 (BafA1) for 1 h. Before cell imaging, cells were treated for 15 min with Magic-red (MR) at 37 °C. Cells were live-imaged by confocal microscopy in the presence of 1 mM HEPES in complete DMEM without phenol red.

To measure the fluorescence intensity ratio between FITC/Alexa Fluor 647-dextran and MR/ Alexa Fluor 647, a threshold was applied to the Alexa Fluor 647-dextran channel to define the lysosomes through the analyse particle macro from ImageJ Fiji software. The background was removed from the obtained fluorescence values.

## Autophagic flux

Cells were treated with 100 nM BafA1 for 2 h. Cell lysates were immunoblotted, and the LC3-II were assessed in cells treated and untreated with BafA1, as previously described (Domingues et al, 2023).

## Hypoxia induction

After 24 h of cell seeding, hypoxia was induced by inserting the plates inside a GasPaK™ EZ – Anaerobe Gas Generating pouch system with an indicator. Cells were exposed to hypoxia conditions for 24 h.

## Cells treatment

For proteasome inhibition, cells were incubated for 4 h with clasto-Lactocystin ß-lactone (10 uM) treatment. MTS1/2 was inhibited by treating the cells with Xmu-mp-1 (1 and 10 μM) for 24 h. For doxorubicin treatment, cells were exposed to 25 μM of doxorubicin.

## Culture and transfection of primary neurons

Cerebrocortical neurons were cultured from E18 to E19 C57BL/6J mice embryos as previously described (Ribeiro et al, 2019), with some minor modifications. Briefly, cortices were washed with ice-cold HBSS three and five times, prior and after papain treatment (20–25 units/mL); prepared in HBSS supplemented with 750 μM EDTA, 1.5 mM CaCl₂,1.6 mM L-Cystein (20 min at 37 °C) treatment, respectively. Cells were mechanically dissociated with a 5 ml glass pipette, no more than 10–15 times with HBSS. After counting, the cells were plated with Neuronal Plating Medium (MEM supplemented with 10% horse serum, 0.6% glucose and 1 mM pyruvic acid) for 2–3 h in 6- or 24-well plates ($92.8 \times 10^3$ cells/cm²) coated with poly-D-lysine (0.1 mg/mL). After this period, the plating medium was removed and replaced by Neurobasal medium supplemented with SM1 supplement (1:50 dilution), 0.5 mM glutamine and 0.12 mg/mL gentamycin. Cultures were fed once a week and maintained in Neurobasal medium supplemented with SM1 supplement and kept in a humidified incubator of 95% air and 5% $CO_2$, at 37 °C, for 12 days.

Neuronal co-transfection was carried out at 10 DIV using 2 μg of plasmid DNA (pFUGW-GFP; a kind gift of Dr. Luis Ribeiro) and 30 pmol of CLN3 scRNA, siRNA13.2, and siRNA13.3 diluted in Tris–EDTA (TE; pH 7.3) and mixed with 2.5 M CaCl₂. This DNA/TE/calcium mix was added to 10 mM HEPES-buffered saline solution (270 mM NaCl, 10 mM KCl, 1.4 mM Na₂HPO₄, 11 mM dextrose, 42 mM HEPES, pH 7.2). Meanwhile, cultured neurons were incubated with cultured conditioned medium with 2 mM kynurenic acid. The precipitates were added drop-wise to each well and incubated at 37 °C/5% $CO_2$, for 1h30. The cells were then washed with HCl-acidified culture medium containing 2 mM kynurenic acid and returned to the 37 °C/5% $CO_2$ incubator for 20–25 min. Finally, the medium was replaced with the initial culture-conditioned medium, and the cells were further incubated in a 37 °C/5% $CO_2$ incubator for 48 h before fixation.

## Statistical analysis

All the experiments were repeated at least three times, using independent experimental samples and statistical tests as specified in the figure legends. The independent replicates were obtained from different passages of the same cell line. Statistical analysis was conducted using Graphpad Prism version 8.0.1. All distributed data are displayed as means ± standard error of the mean (SEM). A *p* value lower than 0.05 was considered statistically significant.

## Data availability

The RNAseq dataset is available at Gene Expression Omnibus under the record GSE265831

The source data of this paper are collected in the following database record: biostudies:S-SCDT-10_1038-S44319-025-00613-3.

## Peer review information

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

## Acknowledgements

We thank Dr. Nicolas Lemus for his expert assistance with RNAseq data normalization and processing, and the CNC microscopy facility. We thank Dr Luis Ribeiro for the reagents and mice primary neuronal cultures preparation, and to Dr Carla Lopes for the optimization of the siRNA concentration of the transfected neuronal cultures. This work was supported by FCT 2022.09311.PTDC, FCT ERC-Portugal, NCL-Stiftung, Welcome Trust Investigator Award in Science 224361/Z/21/Z, La Caixa, John Black Foundation (to IM), the US National Institutes of Health (R01-CA260205 to AB), Italian Telethon Foundation (to AB), Associazione Italiana per la Ricerca sul Cancro A.I.R.C. (IG-22103 to A.B.), European Research Council Advanced grant (H2023 AdG; INCANTAR 101097752 to A.B.). The initial stage of this work was supported by ERC Starting Grant 337327 (to NR) and DFG SFB1190-P02 (to NR and IM). This project also received funding from the European Union's Horizon 2020 research, and innovation program under grant agreement MIA-Portugal No 857524 and the Comissão de Coordenação e Desenvolvimento Regional do Centro—CCDRC through the Centro2020 Program. The content is solely the responsibility of the authors and does not necessarily represent the official views of the National Institutes of Health.

## Author contributions

**Neuza Domingues**: Data curation; Formal analysis; Supervision; Validation; Investigation; Visualization; Methodology; Writing—original draft; Project administration; Writing—review and editing. **Alessia Calcagni'**: Resources; Supervision; Validation; Investigation; Visualization; Methodology; Writing—review and editing. **Sofia Freire**: Validation; Investigation; Visualization; Methodology; Writing—review and editing. **Joana Pires**: Investigation; Writing—review and editing. **Ricardo Casqueiro**: Investigation; Writing—review and editing. **Ivan L Salazar**: Investigation; Methodology; Writing—review and editing. **Niculin Joachim Herz**: Investigation; Writing—review and editing. **Tuong Huynh**: Investigation; Writing—review and editing. **Katarzyna Wieciorek**: Investigation; Writing—review and editing. **Tiago Fleming Outeiro**: Resources; Formal analysis; Supervision; Validation; Visualization; Writing—review and editing. **Henrique Girão**: Formal analysis; Supervision; Validation; Visualization; Writing—review and editing. **Ira Milosevic**: Resources; Software; Formal analysis; Supervision; Validation; Visualization; Writing—review and editing. **Andrea Ballabio**: Resources; Formal analysis; Supervision; Validation; Visualization; Writing—review and editing. **Nuno Raimundo**: Conceptualization; Resources; Data curation; Formal analysis; Supervision;

Funding acquisition; Validation; Visualization; Methodology; Writing—original draft; Writing—review and editing.

Source data underlying figure panels in this paper may have individual authorship assigned. Where available, figure panel/source data authorship is listed in the following database record: biostudies:S-SCDT-10_1038-S44319-025-00613-3.

## Disclosure and competing interests statement

A. Ballabio is cofounder and shareholder of Casma Therapeutics and was an advisory board member of Avilar Therapeutics and Amplify Therapeutics.

# Expanded View Figures

**Figure EV1. Characterization of HEK293T CLN3^KO cell line.** ▶

(A) Schematic representation of the sequence deleted in the CLN3 encoding gene. (B) Putative translation sequences showing the premature stop codons. (C) Representative immunoblot image and quantification of CLN3 protein levels in HEK293T Wt and CLN3^KO cells ($n = 4$). GAPDH was used as loading controls. (D) Confocal fluorescence images of HEK293T Wt and CLN3^KO cells immunostained for Lamp1 protein and the respective quantification of lysosomal area ($n = 3$). Scale bar 10 and 2 μm in the inset images. (E) Quantification of the total number per cell in HEK293T Wt and CLN3^KO cells. At least five cells were quantified per independent experiment ($n = 3$). (F) Representative immunoblot image and quantification of Lamp1 protein levels in Wt and CLN3^KO cells. Ponceau were used as loading controls. (G) Confocal fluorescence images of cells HEK293T CLN3^KO transfected with GFP or CLN3-GFP and immunostained for Lamp1. Quantification of lysosomal size from at least 50 ($n = 3$). (H) Confocal fluorescence images of cells HEK293T parental line transfected with CLN3-GFP and immunostained for Lamp1 ($n = 3$). Scale bar 10 and 2 μm in the inset images. (I) Representative fluorescence images of Wt and CLN3^KO cells loaded with FITC-dextran (Dx), sensitive to pH, and magic red fluorescence, a substrate of cathepsin B. Scale bar 10 μm. The graphs represent the quantification of FITC-dextran (on the top) and magic red fluorescence (on the bottom), normalized to Alexa 647-dextran, not pH sensitive. At least 15 cells were analysed ($n = 3$). The dashed line represents the cell edge. (J) Confocal fluorescence images of cells HEK293T Wt and CLN3^KO immunostained for TFEB. The graph represents the TFEB mean intensity in the nuclei of at least 475 cells ($n = 2$). Yellow dashed lines delineate the nuclei. (K) Representative immunoblot image and quantification of LC3 protein levels in Wt and CLN3^KO cells, untreated and treated with Bafilomycin A1 (BafA1) for 2 h. This blot was also used for the experiment presented in Figure EV3N, and for that reason, the same loading control is used in both panels. Ponceau staining was used as loading controls. Autophagic flux was assessed by the difference between LC3-II levels in BafA1-treated and untreated cells ($n = 4$–5). (L) Differentially expressed genes (DEG) between Wt and CLN3^KO are represented as a volcano plot: log $p$ value adjTtest in the y-axis and log2FC in the x-axis. The transcripts overexpressed in CLN3^KO are to the right of the plot, and on the left, the repressed genes. All the results are mean ± SEM. In the violin plots, the orange dots indicate the mean value of each independent experiment. Statistical differences between the two conditions were assessed by unpaired $t$-test (C, D, H, K-left graph), Sidak's (K, middle graph), or Dunnett's multiple comparisons test (I).

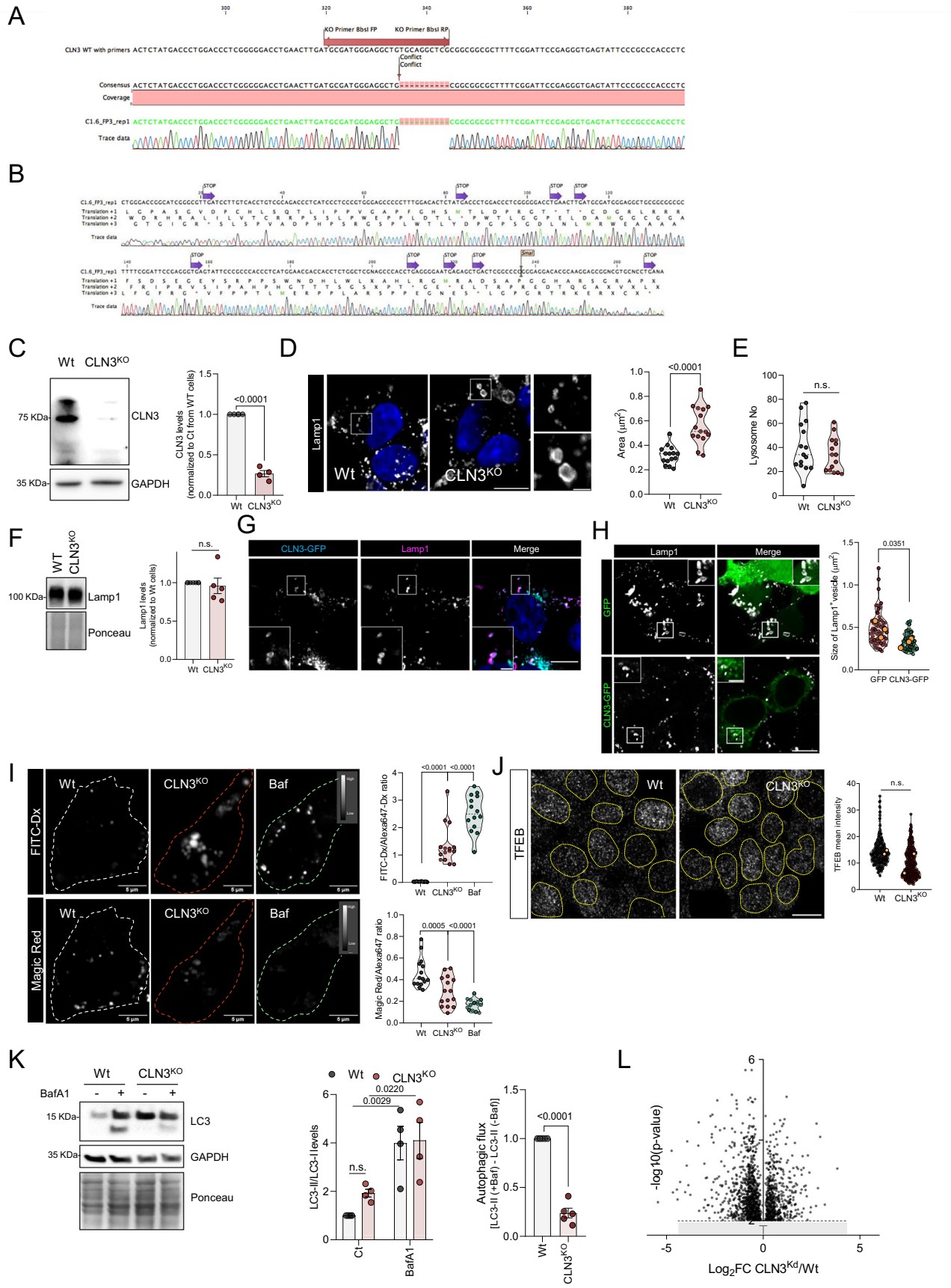

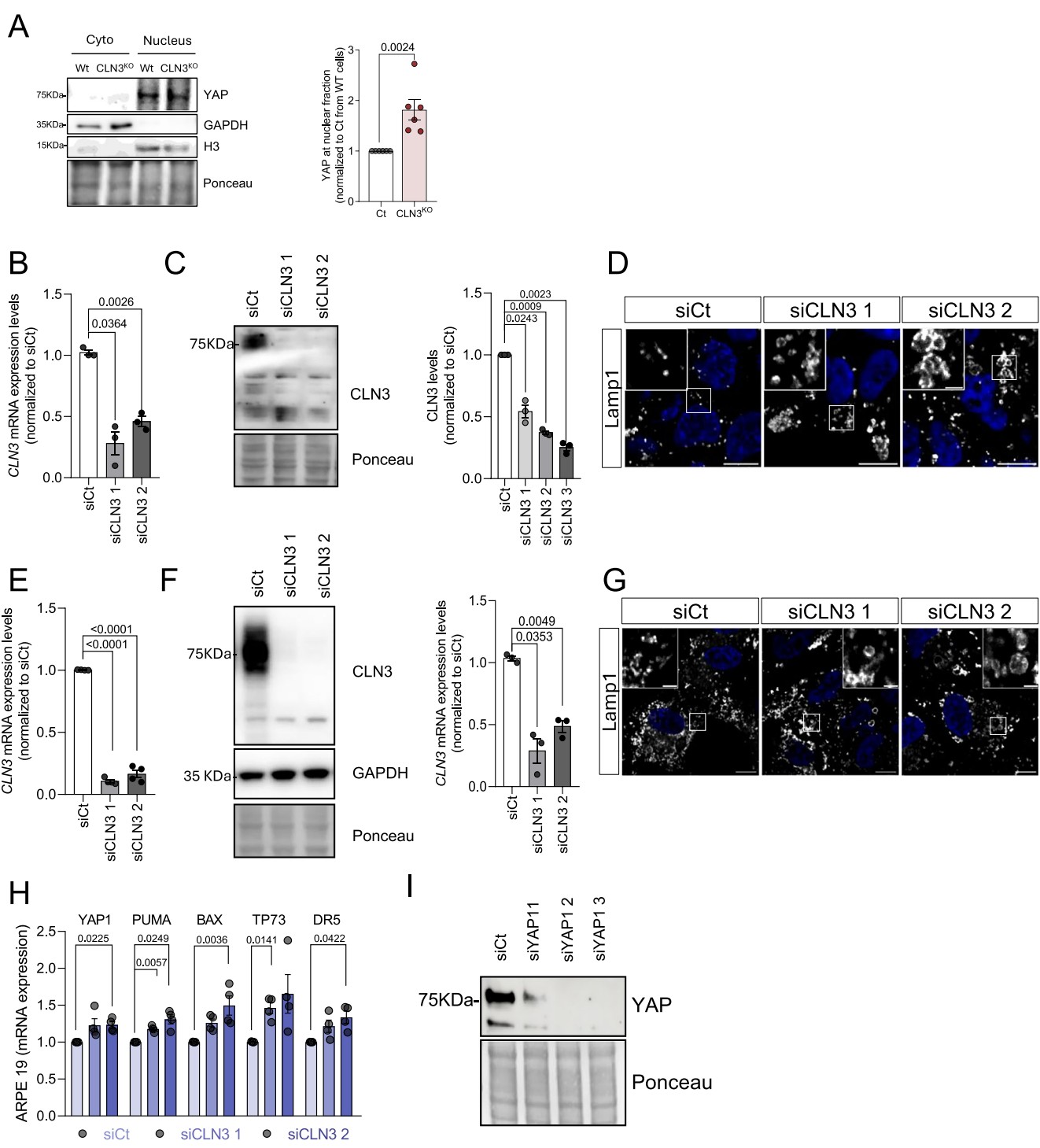

◀ **Figure EV2. Characterization of transient modulation of CLN3 and YAP1 levels in HEK293T parental line cell line.**

(A) Representative immunoblot image and quantification of YAP1 protein levels in the nuclear fraction from Wt and CLN3$^{KO}$ HEK293T cells. Lamin B and GAPDH were used as indicators of nuclear fraction purity ($n = 6$). Ponceau staining was used as loading controls. (B) mRNA levels of the CLN3 gene in HEK293T parental line transfected with siCt or two different siRNAs against CLN3 transcript ($n = 3$). (C) Representative immunoblot image and quantification of CLN3 protein levels in the conditions described in (B) ($n = 3$). Ponceau was used as loading controls. (D) Confocal fluorescence images of cells under the conditions described in (B). Scale bar 10 μm and 2 μm in the inset images. (E) mRNA levels of CLN3 gene in ARPE19 parental line transfected with siCt or two different siRNA against CLN3 transcript ($n = 4$). (F) Representative immunoblot image and quantification of CLN3 protein levels in the conditions described in (E) ($n = 3$). Ponceau was used as loading controls. (G) Confocal fluorescence images of cells under the conditions described in (E). Scale bar 10 and 2 μm in the inset images. (H) mRNA levels of YAP1 target genes involved in apoptosis in ARPE19 parental line transfected with siCt or two different siRNAs against CLN3 transcript ($n = 4$). (I) Representative immunoblot image of YAP1 protein levels from HEK293T parental line transfected with siCt or three different siRNAs against the YAP1 transcript. Ponceau was used as loading controls. All the results are mean ± SEM. Statistical differences between the two conditions were assessed by an unpaired *t*-test (A) and Dunnett's multiple comparisons test (C, E, F, H).

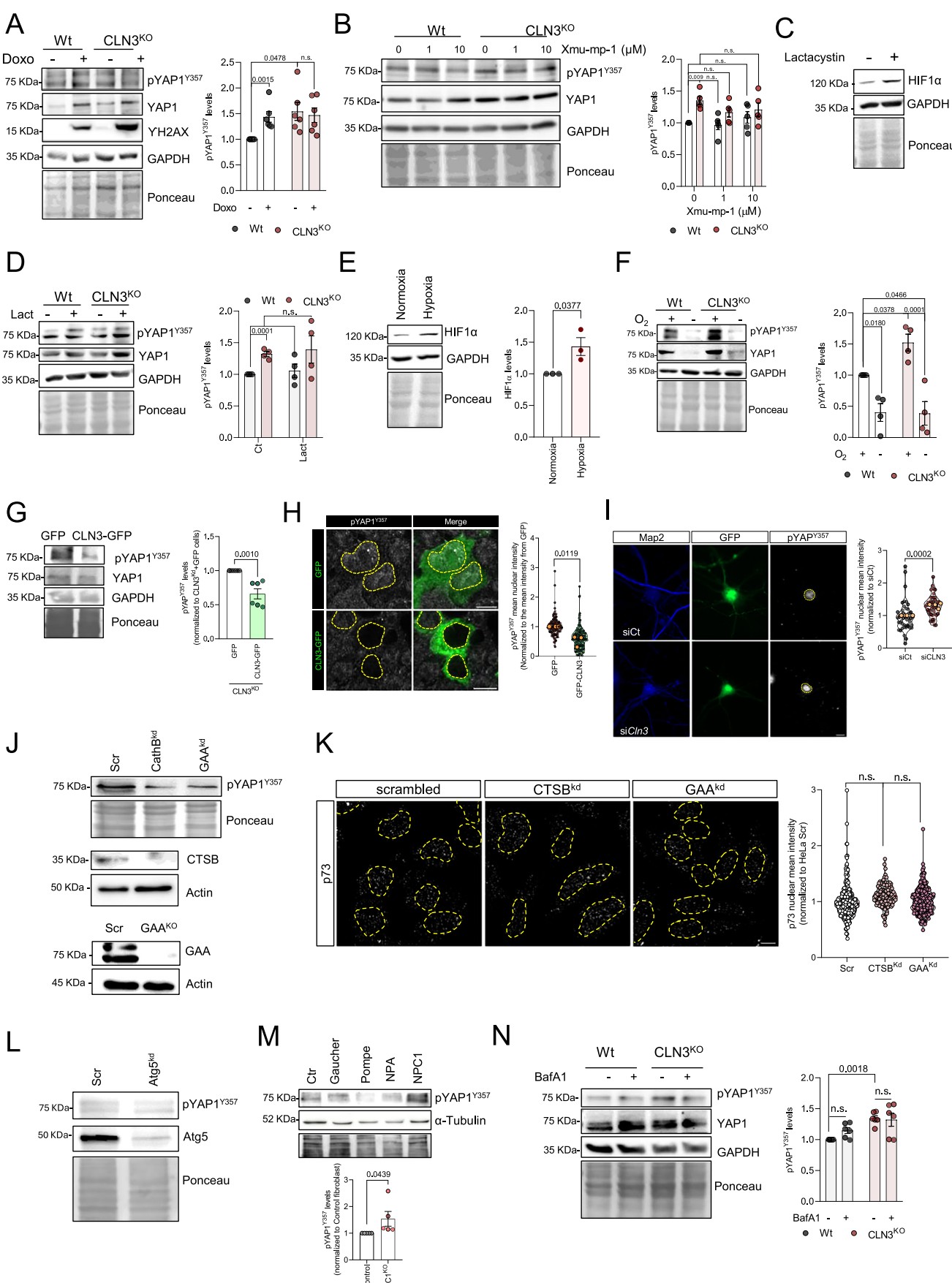

**Figure EV3.   Restoring CLN3 levels in HEK293T CLN3-KO cells led to a decrease in YAP1 phosphorylation at the Tyr357 residue.**

(A, B) Representative immunoblot image and quantification of pYAP1$^{Y357}$ in (A) doxorubicin (Doxo, 25 μM) or (B) Xmu-pu-1-treated (1 and 10 μM) or untreated Wt and CLN3$^{KO}$ cells for 24 h ($n = 4$). γH2Ax immunoblot was used as an experimental control to demonstrate increased DNA damage upon doxorubicin treatment. GAPDH and Ponceau were used as loading controls. (C) Representative immunoblot image of HIF1α in HEK293T Wt cells after Lactocystin (Lact) treatment for 4 h ($n = 4$). HIF1α, a proteasome substrate, was used as a control of proteasome inhibition efficiency. (D) Representative immunoblot image and quantification of pYAP1$^{Y357}$ in Wt and CLN3$^{KO}$ cells in control condition or after Lactocystin (10 μM) treatment in the same conditions as in (C) ($n = 3$). (E) Representative immunoblot image and quantification of HIF1α in Wt in normoxia and hypoxia. GAPDH and Ponceau were used as loading controls ($n = 4$). (F) Representative immunoblot image and quantification of pYAP1$^{Y357}$ (F) in Wt and CLN3$^{KO}$ cells in normoxia and hypoxia ($n = 6$). GAPDH and Ponceau were used as loading controls. (G) Representative immunoblot image and quantification of pYAP1$^{Y357}$ in CLN3$^{KO}$ cells transfected with pEGFP or CLN3-GFP treated. GAPDH and Ponceau were used as loading controls. (H) Confocal fluorescence images and quantification of cells under the conditions described in (G) immunostained for pYAP1$^{Y357}$. Scale bar 10 μm. At least 150 nuclei were analysed ($n = 3$). Nuclei from transfected cells are outlined by the yellow dashed line using Hoechst staining. (I) Confocal fluorescence images and quantification of neurons transfected with siCt or siRNA against CLN3 mRNA, and pEGFP (to label transfected cells) and immunostained for pYAP1$^{Y357}$. Scale bar 10 μm. At least ten nucleus were analysed per independent experiment ($n = 4$). Nuclei are outlined by the yellow dashed line using Hoechst staining. (J) Representative immunoblot image of pYAP1$^{Y357}$ in control, Cathepsin B knockdown (CTSB-Kd) and acid alpha-glucosidase knock-down (GAA-Kd) HeLa cells. Protein depletion was also confirmed by CTSB and GAA immunoblotting. Actin, GAPDH or Ponceau were used as loading controls. Scale bar 10 μm. (K) Confocal fluorescence images and quantification of cells under the conditions described in (J) immunostained for p73. At least a total of 100 nuclei were analysed, 50 in each independent experiment ($n = 2$). Nuclei are outlined by the yellow dashed line using Hoechst staining. Scale bar 10 μm. (L) Representative immunoblot image of pYAP1$^{Y357}$ in control and Atg5 knock-down (Atg5-Kd) HeLa cells. (M) Representative immunoblot image and quantification of pYAP1$^{Y357}$ (from Nieman–Pick, $n = 5$) in human fibroblast from different lysosomal storage diseases: Gaucher, Nieman–Pick (NPC) and Pompe. Tubulin and ponceau were used as loading controls. (N) Representative immunoblot image and quantification of pYAP1$^{Y357}$ in 100 nM of Bafilomycin A1 (BafA1)-treated and untreated Wt and CLN3$^{KO}$ cells for 2 h ($n = 6$). This blot was also used for the experiment presented in Fig. EV1K, and for that reason, the same loading control is used in both panels. GAPDH and Ponceau were used as loading controls. The results are mean ± SEM. In the violin plots, the orange dots indicate the mean value of each independent experiment. Statistical differences between the two conditions were assessed by unpaired *t*-test (**E, D**—between Wt and Wt + Doxo, **G, H, I, M**) and Sidak's multiple comparison test (**A, B, D, F, K, N**).

                                                                    

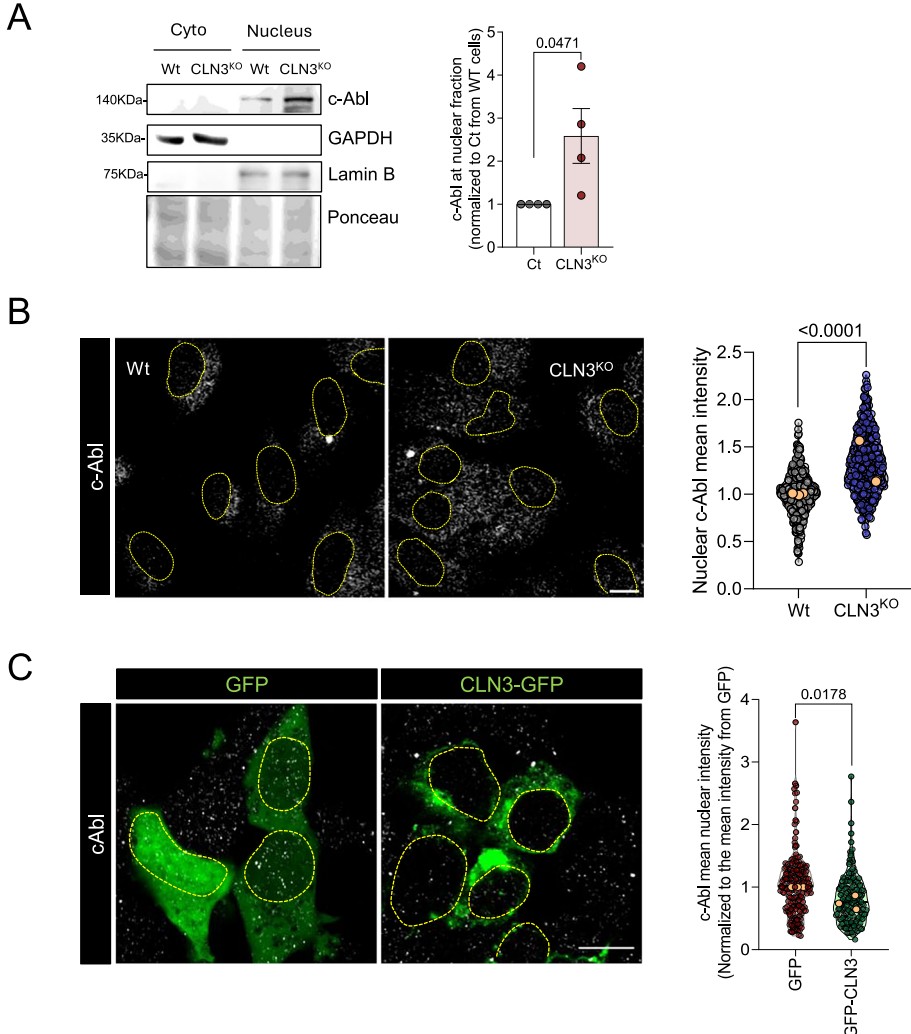

**Figure EV4. Restoring CLN3 levels in HEK293T CLN3-KO cells led to a decrease in c-Abl recruitment to the nucleus.**

(A) Representative immunoblot and quantification of c-Abl in the nuclear fraction of HEK293T Wt and CLN3$^{KO}$ cells ($n = 4$). GAPDH and Lamin B were used as indicators of nuclear fraction purity. (B) Confocal fluorescence images and quantification of ARPE19 Wt and CLN3$^{KO}$ cells immunostained for c-Abl. Nuclei are outlined by the yellow dashed line using Hoechst staining. At least 350 nuclei were analysed ($n = 2$). Scale bar 10 µm. (C) Confocal fluorescence images and quantification of HEK293T CLN3$^{KO}$ cells transfected with GFP or CLN3-GFP and immunostained for c-Abl. At least 150 nuclei were analysed ($n = 3$). Nuclei from transfected cells are outlined by the yellow dashed line using Hoechst staining. Scale bar 10 µm. The results are mean ± SEM. Scale bar 10 µm. Statistical differences between the two conditions were assessed by an unpaired $t$-test. In the violin plots, the orange dots indicate the mean value of each independent experiment.

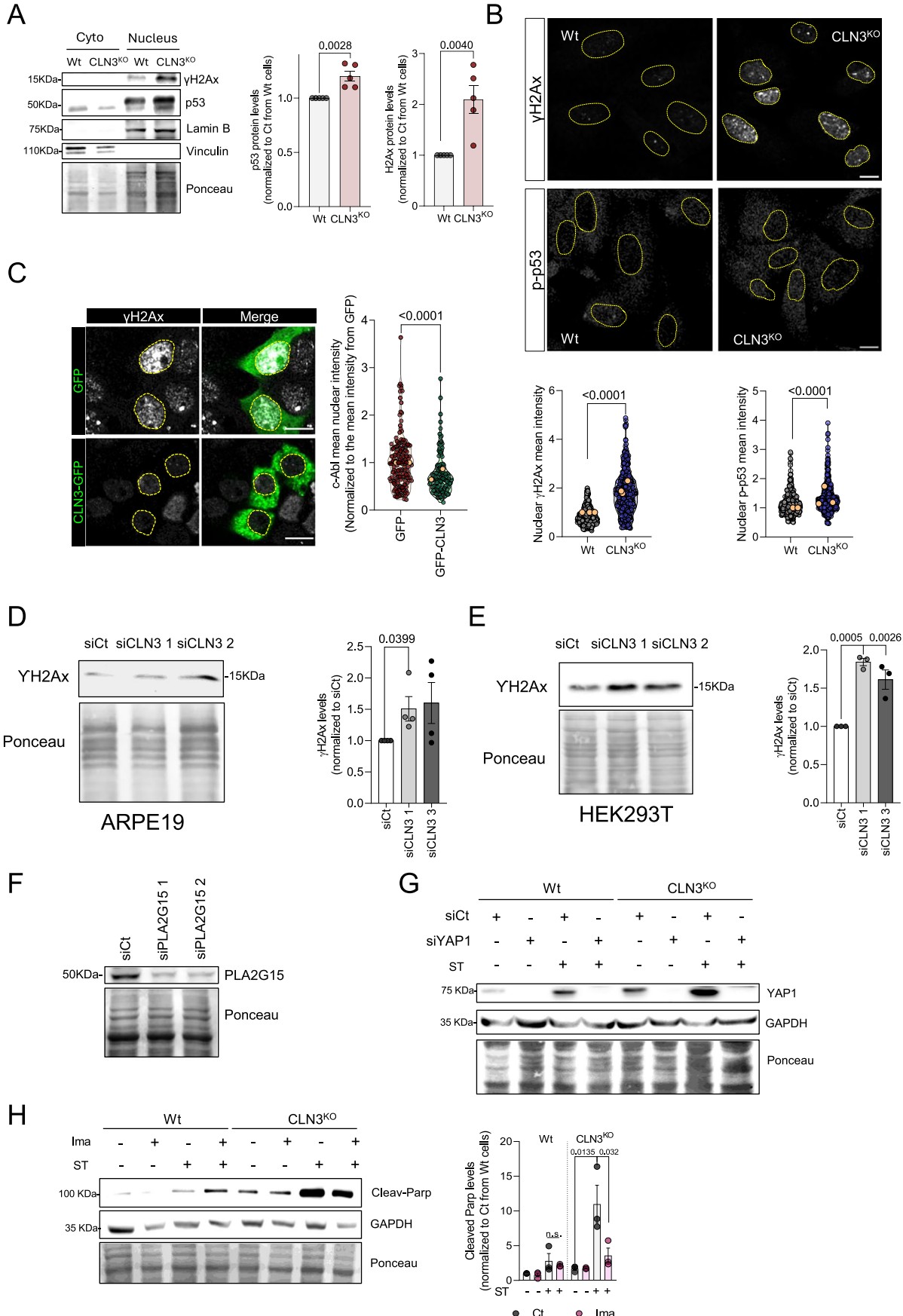

Figure EV5.   Loss of CLN3 function induces an increase in DNA damage and cell cycle arrest, possibly by affecting the nuclear lipidome.

(A) Representative immunoblot image and quantification of p53 and ϒH2Ax in nuclear extracts from HEK293T parental and CLN3-KO cell line ($n = 5$). Ponceau was used as loading controls and vinculin and Lamin B as control for fraction purity. (B) Confocal fluorescence images and quantification of ARPE19 Wt and CLN3$^{KO}$ cells immunostained for ϒH2Ax and p-p53. At least 300 nuclei were analysed ($n = 3$). Nuclei are outlined by the yellow dashed line using Hoechst staining. Scale bar 10 µm. (C) Confocal fluorescence images and quantification of HEK293T CLN3$^{KO}$ cells transfected with GFP or CLN3-GFP, treated and immunostained for ϒH2AX. At least 150 nuclei were analysed ($n = 2$). Scale bar 10 µm. (D, E) Representative immunoblot image of ϒH2Ax in ARPE19 (D, $n = 4$) or HEK293T (E, $n = 3$) parental cell line transfected with siCt or two siRNAs against CLN3 transcripts. Ponceau was used as loading controls. (F) Representative immunoblot image of PLA2G15 in HEK293T parental cell line transfected with siCt or two siRNAs against PLA2G15 transcripts. Ponceau was used as loading controls. (G) Representative immunoblot image and quantification of YAP1 of HEK293T (Wt and CLN3$^{KO}$) cells transfected with siCt or siYAP and treated or untreated with staurosporine (ST) treatment (4 h). GAPDH and ponceau were used as loading controls. (H) Representative immunoblot image and quantification of cleaved PARP of HEK293T (Wt and CLN3$^{KO}$) cells with or without Imatinib incubation for 24 h, and then treated or untreated with staurosporine (ST) (4 h) ($n = 3$). GAPDH and ponceau were used as loading controls. All the results are mean ± SEM. In the violin plots, the orange dots indicate the mean value of each independent experiment. Statistical differences between the two conditions were assessed by using an unpaired $t$-test (A, B, C, H—between CLN3$^{KO}$ cells (Ct and Ima) treated with ST) or Dunnett's multiple comparisons test (D, H). Scale bar 10 µm.

