## [Peer Review File · EMBO Reports]

Loss of the lysosomal protein CLN3 triggers c-Abl-dependent YAP1 pro-apoptotic signaling

Neuza Domingues, Alessia Calcagni, Sofia Freire, Joana Pires, Ricardo Casqueiro, Ivan Salazar, Niculin Herz, Tuong Huynh, Katarzyna Wieciorek, Tiago Outeiro, Henrique Girao, Ira Milosevic, Andrea Ballabio, and Nuno Raimundo

Corresponding author(s): Nuno Raimundo (nuno.raimundo@psu.edu), Neuza Domingues (neuza.domingues@uc.pt)

Review Timeline:

Transfer Date:	15th Apr 25
Editorial Decision:	28th Apr 25
Revision Received:	17th Jul 25
Editorial Decision:	1st Sep 25
Revision Received:	17th Sep 25
Accepted:	2nd Oct 25

Transaction Report: This manuscript was transferred to EMBO reports following peer review at The EMBO Journal.

Referee #1:

No further comment all my concerns have been addressed, based on the novelty of the work I would recommend acceptance, I would also encourage the editors to consider a commentary.

Referee #2:

Considering the fact that the cAbl-YAP-p73 pathway induce apoptosis is well known in the field, the impact of this study remains modest without showing how the lack of CLN3 activates DNA damage.

Specific:

The authors proposed to defer showing the molecular mechanism as to how downregulation of CLN3 leads to DNA damage (and c-Abl activation) to future publications.

The quality of the data is modest. One example is that CLN3 downregulation obviously increases the level of YAP. Thus, it is likely that increases in YAP phosphorylation at Y357 can take place even without increases in c-Abl.

The quality of the data in many experiments, including Figure 2A S2A, and S3A, are very modest.

Despite this reviewer's request to investigate how CLN3 affects Hippo kinases, the authors conduct only hypoxia experiments in S3E without showing data regarding Hippo kinases.

The authors' conclusion as to CLN3 KO directly affects c-Abl independently of lysosomal inactivation is weakly supported by the data shown in Fig. 3M.

The role of the YAP-p73 pathway in mediating CLN3KO's effect upon apoptosis is tested only in vitro and became obvious only in the presence of staurosporin.

What is the significance of YAP phosphorylation at Y357 in overall cellular effects of CLN3 downregulation? The data shown in 4K is quite modest, what is the effect of potent inhibition of autophagy?

The data quality of the study is modest: critical controls, such as control IP data and wild type animal data are missing in some experiments.

Dear Dr. Raimundo

Thank you for the transfer of your manuscript to EMBO Reports. Your study had been re-reviewed after revision at our sister journal The EMBO Journal and given the remaining concerns raised by referee #2 the editors could not offer publication.

Given the potential interest of your findings and the positive evaluation by referee #1 we have invited you to revise your study for potential publication in EMBO Reports, under the condition that the following points from referee #2 can be addressed satisfactorily:

- (Specific) Point 3: I agree with the referee regarding this point and note that the Western blots in question have a rather high contrast modification. Please reduce brightness/contrast and repeat these experiments.
 - Point 4: I agree that the data on the involvement of MST1/2 are correlative and suggest testing the contribution of HIPPO pathway kinases more directly using e.g., inhibitors or siRNA.
 - Point 5: Please address these concerns by repeating the experiments and by quantifying more samples to strengthen these conclusions.
 - Point 8: Please ensure that all relevant controls are provided.
- Point 2, 6: please discuss the limitations or alternative scenarios noted by the referee.

Please address these remaining concerns also in a point-by-point response.

=====

In addition to these scientific concerns, there are also a number of things we need from the editorial side:

- Your study will be considered as Report, which requires the combination of Results & Discussion.
- Please provide up to 5 keywords.
- The manuscript sections are in the wrong order. Please order them like this:
Title page - Abstract - Introduction - Results & Discussion - Methods - Acknowledgements - Disclosure and competing interests statement - References - Figure legends - Tables and their legends (not EV tables) - Expanded View Figure legends
- Please write the Abstract in present tense.
- Please complete the funding information in the online manuscript tracking system. All funding sources mentioned in the Acknowledgments should also be in our system, with project numbers where available.
- Regarding the Author Contributions, we now use CRediT to specify the contributions of each author in the journal submission system. Therefore, please remove the Author Contributions from the manuscript file and make sure that the author contributions in our online manuscript tracking system are correct and up-to-date. The information you specified in the system will be automatically retrieved and typeset into the article. You can enter additional information in the free text box provided, if you wish.
- Please rename the 'Potential Conflicts of interest' paragraph to 'Disclosure and competing interests statement'. For more information see <https://www.embopress.org/page/journal/14693178/authorguide#conflictsofinterest>
- Appendix: please remove the supplementary methods and move them to the main manuscript text.
- Please provide a Title page with a table of contents incl. page numbers.
- The correct nomenclature is "Appendix Figure S1"etc and "Appendix Table S1"
- Please download and fill our Reagents and Tools Table template (.docx), which you can find in our author guidelines: <https://www.embopress.org/page/journal/14693178/authorguide#structuredmethods>.
When submitting your revised manuscript, please do not include the Reagents and Tools Table in the Methods section of the manuscript but upload it as a separate file choosing the file type "Reagent Table".
An example of a Method paper with Structured Methods can be found here: <https://www.embopress.org/doi/10.15252/msb.20178071>.
- There is a callout for a Table S2 in the methods section, which is absent. Please correct the callout. Please also provide a callout for Fig 3F, wherever appropriate.

- The correct order of the manuscript sections should be as follows: Abstract, Keywords, Introduction, Results & Discussion, Methods, Acknowledgements, Disclosure and competing interests statement, References, Figure legends, Expanded View Figure legends
- The main figures appear to have rather low resolution. Please provide files that have higher resolution and production quality.
- We perform a routine image integrity check on all revised manuscripts. In this case we noticed that the GAPDH blots shown in Figure S1K and Figure S3M appear to be the same. Please check these figure panels, clarify and ensure that the appropriate controls are shown for each Western blot/experiment.
- Source Data: I did a spot check and the source data for Figure 1D as well as for Figure 2B seemed not to match. But this was difficult to assess, due to the limited resolution of the figure files. Please make sure that all source data are correct.
- We perform a data integrity check on all quantitative source data files and noticed the following aberrations that need clarification:
 - Source Data xls file for Fig 1K: the values in row 5 (BAX) and row 7 (DR5) are exactly the same. This is also reflected in the figure panel. Please check.
 - Source Data xls file for Fig. 1G: The values for PUMA match exactly with those for DR5 in the .xls file. The values seem also to be the same in the figure panel (bars, error bars and data points are the same).
 - Source data xls file for Fig. 2D: values for Hippocampus and Thalamus are identical. Please check and clarify.
 - Source data xls file for Fig. 2H: Column B, rows 287 - 323 are repeated as block in Column B, rows 609 - 645. Those two rows of numbers are identical to each other, i.e., B287 = B609, B288 = B610 etc. Please check and clarify.
 - Source data .xls file for Fig. 3E: blocks of measurements seem to have been repeated and duplicated between CLN3-KO and lma. I attach the color-coded file. If you look at the green numbers, e.g., 1,01192 and then the series of measurements that follows beneath, you will see that the numbers match between CLN3-KO and lma. If you start again at the green 0,90244 the same pattern emerges with all measurements in the following rows being equal between both conditions. Please check these measurements and please clarify these duplication.
- Our production/data editors have asked you to clarify several points in the figure legends (see below). Please incorporate these changes in the manuscript and return the revised file with tracked changes with your final manuscript submission.

A) Figure legend text:

1. Please note that the legends for supplementary figures 1G, H is not provided in the sequential manner (legend for supplementary figure 1H is provided before legend of supplementary figure 1G). This needs to be rectified.

B) Statistical test information. Only p-values that are actually shown in the figure panel(s) should (and must) be defined in the legends, all others should be removed from (or added to) the legend. Moreover, we ask for the specification of exact p-values:

1. Please note that the exact p values are not provided in the legends of figures 1D-H, I-J, L; 2A, B, D, E, F, H, I, J; 3A-F; 4B, C, E-K; supplementary figure(s) 1C, D, H, I, K; 2A, B, C, E, F, H; 3A, C, D, E, F, G, H, J, L, M; 4A, B, C; 5A, C, D, E, G.

2. Please indicate the statistical test used for data analysis in the legends of figures 1B, C.

3. Please note that in figures 1I, J; supplementary figure(s) 1K, there is a mismatch between the annotated p values in the figure legend and the annotated p values in the figure file that should be corrected.

C) Replicates and error bars:

1. Please note that the box plots need to be defined in terms of minima, maxima, centre, bounds of box and whiskers, and percentile in the legend of figure 1D.

2. Please note that the error bars are not defined in the legends of figures 1E, F, G, H, I, J, L.

D) Data presentation:

1. Please note that the scale bar needs to be defined for figures 4C, F, H; supplementary figure(s) 3J, 4B, C; 5B, C.

2. Please note that scale bar and its definition are missing for figure 2D.

3. Please note that the yellow dashed border is not defined in the legend of supplementary figures 3G, H, J; This needs to be rectified.

- Finally, EMBO Reports papers are accompanied online by

- A) a short (1-2 sentences) summary of the findings and their significance,
- B) 2-3 bullet points highlighting key results and

C) a schematic summary figure that provides a sketch of the major findings (not a data image). Please provide the summary figure as a separate file in PNG or JPG format at a size of 550x300-600 pixels (width x height). Please note that the size is rather small and that text needs to be readable at the final size. Please send us this information along with the revised manuscript.

With kind regards,

We thank the Editor and Referee #1 for highlighting the interest of our findings. We have performed additional experiments to address the comments from Referee #2, as detailed below.

Point 3: I agree with the referee regarding this point and note that western blots in question have a rather high contrast modification. Please reduce brightness/contrast and repeat these experiments

We have now performed additional repetitions for the western blot from Fig2A, to attempt to obtain a less contrasted image. Accordingly, we have changed the panel from HEK293 cells at the main figures Fig2A (also shown here in Rebuttal Figure 1). For ARPE19 cell lines, we performed further repetitions of the western blot (as shown here in Rebuttal Figure 1), but the ARPE19 cells western blot presented at the Fig2A was still the most representative of the quantification, so we kept it while reducing the contrast in all proteins. We also reduced the contrast of images in **Figure EV2A** and **EV3A**. Additionally, we have increased the replicates for S3A.

Rebuttal Figure 1 - Representative immunoblot image and quantification of YAP1 protein levels phosphorylated at the tyrosine residue 357 (pYAP1^{Y357}) in HEK293 and ARPE19 Wt and CLN3^{KO} cells. GAPDH and Ponceau were used as loading controls.

It is noteworthy that pYAP1^{Y357} and YAP1 immunoblots are known to show cell-specificity and to present less defined bands in some cell lines (see for example Levy et al., 2008, Mol Cell. 29: 350-61, PMID18280240, as illustrated below in Rebuttal Figure 2). For example: the pYAP^{Y357} band in ARPE19 (Fig2a), is much more defined than the bands observed bands in HEK293 cells, despite these are both human cell lines and the same whole cell extract protocol was used.

Rebuttal Figure 2

REDACTED: Figure 3C, D from Levy *et al*, 2008
doi: 10.1016/j.molcel.2007.12.022.

Rebuttal Figure 2 - Immunoblot image and quantification of YAP1 protein levels phosphorylated at the tyrosine residue 357 (pYAP1^{Y357}) from Levy et al work (Levy *et al*, 2008).

Point 4: I agree that the data on the involvement of MST1/2 are correlative and suggest testing the contribution of HIPPO pathway kinases more directly using e.g, inhibitors or siRNA.

We have now performed experiments using Xmu-mp-1, a reversible and selective MST1/2 inhibitor (Fan et al, 2016)(Fan et al, 2016). These results are now included in Fig S3B.

We observed that inhibition of MTS1/2 and the consequent reduction of pYAP^{S127} (Shalhout et al, 2021; Han et al, 2022)(Shalhout et al, 2021; Han et al, 2022), as observed in Wt and CLN3-KO HEK293 cells treated with Xmu-mp-1 (Figure 3A), did not affected the levels of pYAP^{Y357} (Figure 3B). We have also previously shown that activation of Hippo pathway by hypoxia did not affect the levels of pYAP^{Y357}. We had previously assessed the total levels pYAP^{Ser127}, which were not changed at CLN3-KO cells (Fig2C), and measured pYAP^{Y357} in conditions of proteasome inhibition (Figure EV3C), without significant alterations. Therefore, these new results obtained under conditions of MST1/2 inhibition are coherent with our previous findings presented in the revised version of this manuscript: in cells lacking CLN3, the pathway driving YAP1 signaling is dependent only on activation of c-Abl and not on the canonical Hippo pathway.

When Hippo pathway is ON, MST1/2 phosphorylate Lats1/2, which is responsible to phosphorylate YAP1 at Ser127, increasing the cytosolic retention of YAP1 and its proteasomal degradation. Our results show that this pathway is not activated in CLN3-KO cells, suggesting that Hippo pathway is not involved in pYAP^{Y357} signalling cascade in CLN3-KO cells. In Figure 3B, we present the immunoblot of H2Ax showing no alteration of DNA damage induced by Xmu-mp-1, which is in agreement with the unchanged levels of pYAP^{Y357}. As we previously discussed, the pYAP^{Y357}-dependent pathway is only activated under conditions of DNA damage, as in CLN3-KO or control cells treated with an alkylating agent (Figure EV3A).

Rebuttal Figure 3 - Representative immunoblot image and quantification of YAP1 protein levels phosphorylated at the serine127 (pYAP1^{S127}) and tyrosine residue 357 (pYAP1^{Y357}) in HEK293 untreated or treated with 1 or 10 μM of Xmu-mp-1 for 24h. GAPDH and Ponceau were used as loading controls.

Point 5: Please address these concerns by repeating the experiments and by quantifying more samples to strengthen these conclusions

R#2. "The authors' conclusion as to CLN3 KO directly affects c-Abl independently of lysosomal inactivation is weakly supported by the data shown in Fig. 3M."

To test if the activation of c-Abl observed in CLN3-KO was downstream of lysosomal perturbations, we assessed the levels of pYAP^{Y357} in human fibroblast from patients with different lysosomal storage disease (**Figure EV3M** from this new version of the manuscript), well known to present defects on lysosomal acidification (e.g. (Yambire *et al*, 2019a, 2019b; Agostini *et al*, 2024)). We had also included at the revised version the experiments including Wt and CLN3-KO cells exposed for 24h to BafA1, a well established inhibitor of lysosomal activity (Fig. S3M from the R1 version of the manuscript)

We have now repeated this experiment, and the data is now included at **Figure EV3N** of the new version of this manuscript.

Point 8: Please ensure that all relevant controls are provided.

R#2. The data quality of the study is modest: critical controls, such as control IP data and wild type animal data are missing in some experiments.

We had included the negative control for the IP no YAP1 antibody (added an IgG from the same species of the antibody – rabbit).

All the data acquired from the animal model for Batten's disease were obtained using simultaneously control animals, as indicated in the text, figures and methods.

Point 2, 6: please discuss the limitation or alternative scenarios noted by the referee

R#2. "The quality of the data is modest. One example is that CLN3 downregulation obviously increases the level of YAP. Thus, it is likely that increases in YAP phosphorylation at Y357 can take place even without increases in c-Abl."

Increased levels of pYAP1^{Y357} protein were only shown in DNA damage conditions (as the one reported in our work). In addition, Levy et al demonstrated "endogenous Yap1 protein is elevated upon cisplatin treatment Levy et al, 2008; Figure 7b) (Levy et al, 2008), however, the molecular mechanism for it is not understood yet. Nevertheless, increased basal levels of YAP1 will not force the c-Abl activity without an additional stress condition responsible for c-Abl activation. Accordingly, in Levy et al work, constitutive active c-Abl immunoprecipitated with pYAP1^{Y357}, (Figure 4, which represent a crop from Fig 2c of Levy et al work), and this phosphorylation is stabilized by c-Abl under DNA damage condition. Thus, independently of the basal YAP1 levels increased in conditions of CLN3-KO conditions (=DNA damage), they will not represent an increase in pYAP1^{Y357}, if c-Abl was not previously activated. Finally, it is this YAP1 phosphorylation that is responsible for the pro-apoptotic phenotype independently of total YAP1 levels, showing only a reduction in conditions of c-Abl inhibition in CLN3-KO cells, as shown with the imatinib experiments.

Rebuttal Figure 4

REDACTED: Figure 2C from Levy D et al, 2008
doi: 10.1016/j.molcel.2007.12.022

R#2. 6 The role of the YAP-p73 pathway in mediating CLN3KO's effect upon apoptosis is tested only in vitro and became obvious only in the presence of staurosporin.

We show here that the CLN3-KO cells are predisposed to apoptotic death. These cells are culture in the perfect growth conditions, in the presence of serum which is known to have anti-apoptotic effects. Therefore, a stimulus is needed to induce apoptotic cell death. The fact that when subject to staurosporine treatment, a known inducer of apoptosis, the CLN3-KO cells die more than the WT shows that they are more susceptible. We observed a similar effect when we treated mitochondrial A1555G cybrids with staurosporine (Raimundo *et al*, 2012). Of note, we also verified that this cell line exhibits cell cycle arrest and increased DNA damage, which are features of a pro-apoptotic phenotype.

While the cell death experiment was made *in vitro*, we showed that the brain of the CLN3 mouse model also presented similar markers regarding the YAP1 pathway. R#2 chose to ignore that. Nevertheless, we would like to acknowledge the R#2 for recognition that the effect with staurosporine was obvious.

The possible additional effects of PLA2G15 were addressed in the discussion, as requested.

We have addressed above all the points requested by the Editor.

References

- Agostini F, Pereyra L, Dale J, Yambire KF, Maglioni S, Schiavi A, Ventura N, Milosevic I & Raimundo N (2024) Upregulation of cholesterol synthesis by lysosomal defects requires a functional mitochondrial respiratory chain. *Journal of Biological Chemistry* 300: 107403
- Fan F, He Z, Kong LL, Chen Q, Yuan Q, Zhang S, Ye J, Liu H, Sun X, Geng J, *et al* (2016) Pharmacological targeting of kinases MST1 and MST2 augments tissue repair and regeneration. *Sci Transl Med* 8
- Han H, Nakaoka HJ, Hofmann L, Zhou JJ, Yu C, Zeng L, Nan J, Seo G, Vargas RE, Yang B, *et al* (2022) The Hippo pathway kinases LATS1 and LATS2 attenuate cellular responses to heavy metals through phosphorylating MTF1. *Nature Cell Biology* 2022 24:1 24: 74–87
- Levy D, Adamovich Y, Reuven N & Shaul Y (2008) Yap1 Phosphorylation by c-Abl Is a Critical Step in Selective Activation of Proapoptotic Genes in Response to DNA Damage. *Mol Cell* 29: 350–361

Raimundo N, Song L, Shutt TE, McKay SE, Cotney J, Guan MX, Gilliland TC, Hohuan D, Santos-Sacchi J & Shadel GS (2012) Mitochondrial stress engages E2F1 apoptotic signaling to cause deafness. *Cell* 148

Shalhout SZ, Yang PY, Grzelak EM, Nutsch K, Shao S, Zambaldo C, Iaconelli J, Ibrahim L, Stanton C, Chadwick SR, *et al* (2021) YAP-dependent proliferation by a small molecule targeting annexin A2. *Nature Chemical Biology* 2021 17:7 17: 767–775

Yambire KF, Mosquera LFe, Steinfeld R, Muhle, Chhristiane;, Elina Ikonen;, Milosevic I & Raimundo N (2019a) Mitochondrial biogenesis is transcriptionally repressed in lysosomal lipid storage diseases. *Elife* 8: e39598

Yambire KF, Rostosky C, Watanabe T, Pacheu-Grau D, Torres-Odio S, Sanchez-Guerrero A, Senderovich O, Meyron-Holtz EG, Milosevic I, Frahm J, *et al* (2019b) Impaired lysosomal acidification triggers iron deficiency and inflammation in vivo. *Elife* 8: e51031

Dear Nuno,

Thank you for the submission of your revised manuscript to EMBO Reports. As you will see from the report below, referee #2 considers the remaining concerns adequately addressed and recommends publication.

Before we can proceed with the official acceptance of your manuscript, I kindly ask you to address the following requests from the editorial side. These were already part of my last decision letter, but you might have overlooked them:

- 1) Your study will be considered as Report, which requires the combination of Results & Discussion.
- 2) Please provide up to 5 keywords on the title page of the manuscript.
- 3) The correct order of the manuscript sections should be as follows: Abstract, Keywords, Introduction, Results & Discussion, Methods, Acknowledgements, Disclosure and competing interests statement, References, Figure legends, Expanded View Figure legends
- 4) As a standard procedure we modify title and abstract. Please see my suggestion below my signature.
- 5) Please complete the funding information in the online manuscript tracking system. All funding sources mentioned in the Acknowledgments should also be in our system, with project numbers where available.
- 6) Regarding the Author Contributions, we now use CRediT to specify the contributions of each author in the journal submission system. Therefore, please remove the Author Contributions from the manuscript file and make sure that the author contributions in our online manuscript tracking system are correct and up-to-date. The information you specified in the system will be automatically retrieved and typeset into the article. You can enter additional information in the free text box provided, if you wish.
- 7) Please rename the 'Potential Conflicts of interest' paragraph to 'Disclosure and competing interests statement'. For more information see <https://www.embopress.org/page/journal/14693178/authorguide#conflictsofinterest>
- 8) The main figures appear to have rather low resolution. When I zoom in, the images and text quickly become pixelated. Please provide files that have higher resolution and production quality.
- 9) Supplementary information file:
 - A) Figure EV1 to EV5 should be provided as individual production quality figure files. Their legends are part of the main manuscript in a section called Expanded View Figure legends, placed after the main figure legends. The figures need to fit on one page. If needed, you can split one of the figures in 2 figures.
 - B) Please remove the supplementary methods and move them to the main manuscript text.
 - C) The remaining Table 1 could be uploaded as Table EV1 (please update the callouts in the manuscript.)
- 10) Callouts that also need correction: Supplementary Table S1, Supplementary Figure 5F
- 11) Please download and fill our Reagents and Tools Table template (.docx), which you can find in our author guidelines: <https://www.embopress.org/page/journal/14693178/authorguide#structuredmethods>. When submitting your revised manuscript, please do not include the Reagents and Tools Table in the Methods section of the manuscript but upload it as a separate file choosing the file type "Reagent Table". An example of a Method paper with Structured Methods can be found here: <https://www.embopress.org/doi/10.15252/msb.20178071>.
- 12) There is a callout for a Table S2 in the methods section, but such a table is not present. Please check and correct the callout.
- 13) We perform a routine image integrity check on all revised manuscripts. In this case we noticed that the GAPDH blots shown in Figure EV1K and Figure EV3N appear to be the same. Please check these figure panels, clarify and ensure that the appropriate controls are shown for each Western blot/experiment. Are the Western blots shown in these two panels and their controls from the same experiment?
- 14) Source Data: I did a spot check and the source data for Figure 1D seemed not to match. I could not find the cells shown in the figure panels in the source data image. But this was difficult to assess, due to the limited resolution of the figure files. Please make sure that all source data are correct.

15) Source Data: In my previous decision letter I told you that we perform a data integrity check on all quantitative source data files and that I had noticed the following aberrations that needed clarification (a-e). I checked and noticed that you have updated the source data as follows (indicated with =>):

a) Source Data xls file for Fig 1K: the values in row 5 (BAX) and row 7 (DR5) are exactly the same. This is also reflected in the figure panel. Please check.

=> You have replaced the data for DR5.

b) Source Data xls file for Fig. 1G: The values for PUMA match exactly with those for DR5 in the .xls file. The values seem also to be the same in the figure panel (bars, error bars and data points are the same).

=> You have replaced the data for DR5.

c) Source data xls file for Fig. 2D: values for Hippocampus and Thalamus are identical. Please check and clarify.

=> You have replaced the Hippocampus data.

d) Source data xls file for Fig.2H: Column B, rows 287 - 323 are repeated as block in Column B, rows 609 - 645. Those two rows of numbers are identical to each other, i.e., B287 = B609, B288 = B610 etc. Please check and clarify.

=> It appears that the entire .xls file and quantification has been replaced as all numbers are different from the previous version.

e) Source data .xls file for Fig. 3E: blocks of measurements seem to have been repeated and duplicated between CLN3-KO and Ima. I attach the color-coded file. If you look at the green numbers, e.g., 1,01192 and then the series of measurements that follows beneath, you will see that the numbers match between CLN3-KO and Ima. If you start again at the green 0,90244 the same pattern emerges with all measurements in the following rows being equal between both conditions. Please check these measurements and please clarify these duplication.

=> The file has been updated, which resulted in a significant change in the violin blot for Ct and Ima shown in Fig. 3E.

I appreciate that you have corrected the discrepant data but could you please add a short note explaining the earlier detected duplications and the changes I noted for Figure 2H and the changes to the quantification results now shown in Figure 3E?

Thank you very much.

16) Our production/data editors have asked you to clarify several points in the figure legends (see below). Please incorporate these changes in the manuscript and return the revised file with tracked changes with your final manuscript submission.

=> Please note that this analysis was done on the previous version of your manuscript, but the requests have not been implemented. The figure panels might have been changed to some extent, so please keep in mind these general specifications:

(a) quantifications need information on the number of repeats (technical or biological), the statistical test used, the exact p-values unless $p < 0.0001$, a description of bars and error bars, violin blots or box plots, individual data points need to be added, (b) imaging data need scale bars and any labeling in the image (e.g., arrows or dashed lines) must be defined in the legend (the latter also applies to labels on Western blots).

The requests on the previous version were the following:

A) Figure legend text:

1. Please note that the legends for supplementary figures 1G, H is not provided in the sequential manner (legend for supplementary figure 1H is provided before legend of supplementary figure 1G). This needs to be rectified.

B) Statistical test information. Only p-values that are actually shown in the figure panel(s) should (and must) be defined in the legends, all others should be removed from (or added to) the legend. Moreover, we ask for the specification of exact p-values:

1. Please note that the exact p values are not provided in the legends of figures 1D-H, I-J, L; 2A, B, D, E, F, H, I, J; 3A-F; 4B, C, E-K; supplementary figure(s) 1C, D, H, I, K; 2A, B, C, E, F, H; 3A, C, D, E, F, G, H, J, L, M; 4A, B, C; 5A, C, D, E, G.

2. Please indicate the statistical test used for data analysis in the legends of figures 1B, C.

3. Please note that in figures 1I, J; supplementary figure(s) 1K, there is a mismatch between the annotated p values in the figure legend and the annotated p values in the figure file that should be corrected.

C) Replicates and error bars:

1. Please note that the box plots need to be defined in terms of minima, maxima, centre, bounds of box and whiskers, and percentile in the legend of figure 1D.

2. Please note that the error bars are not defined in the legends of figures 1E, F, G, H, I, J, L.

D) Data presentation:

1. Please note that the scale bar needs to be defined for figures 4C, F, H; supplementary figure(s) 3J, 4B, C; 5B, C.

2. Please note that scale bar and its definition are missing for figure 2D.

3. Please note that the yellow dashed border is not defined in the legend of supplementary figures 3G, H, J; This needs to be rectified.

17) Finally, EMBO Reports papers are accompanied online by

A) a short (1-2 sentences) summary of the findings and their significance,

B) 2-3 bullet points highlighting key results and

C) a schematic summary figure that provides a sketch of the major findings (not a data image). Please provide the summary figure as a separate file in PNG or JPG format at a size of 550x300-600 pixels (width x height). Please note that the size is rather small and that text needs to be readable at the final size. Please send us this information along with the revised manuscript.

With kind regards,

Martina

Referee #1:

The authors are responsive and the paper is improved. The findings are novel and the impact is high.

Abstract

Batten disease is characterized by early-onset blindness, juvenile dementia and death within the second decade of life. The most common genetic cause are mutations in CLN3, encoding a lysosomal protein. Currently, no therapies targeting disease progression are available, largely because its molecular mechanisms remain poorly understood. To understand how CLN3 loss affects cellular signaling, we generated human CLN3 knock-out cells (CLN3-KO) and performed RNA-seq analysis. Our multi-dimensional analysis reveals the transcriptional regulator YAP1 as a key factor in remodeling the transcriptome in CLN3-KO cells. YAP1-mediated pro-apoptotic signaling is increased as a consequence of CLN3 functional loss in retinal pigment epithelia cells, and in the hippocampus and thalamus of CLN3 Δ 7/8 mice, an established model of Batten disease. Loss of CLN3 leads to DNA damage, activating the kinase c-Abl which phosphorylates YAP1, stimulating its pro-apoptotic signaling. This novel molecular mechanism underlying the loss of CLN3 in mammalian cells and tissues may pave a way for novel c-Abl-centric therapeutic strategies to target Batten disease.

1) Your study will be considered as Report, which requires the combination of Results & Discussion.

2) Please provide up to 5 keywords on the title page of the manuscript.
Lysosomes, CLN3, Batten disease, lysosome-nucleus communication, YAP1, DNA damage

3) The correct order of the manuscript sections should be as follows: Abstract, Keywords, Introduction, Results & Discussion, Methods, Acknowledgements, Disclosure and competing interests statement, References, Figure legends, Expanded View Figure legends

Our manuscript section follows now the requested order.

4) As a standard procedure we modify title and abstract. Please see my suggestion below my signature.

We apologize, but we did not find the title suggestion in the e-mail send It by the Editor.

Abstract as modified by the editor was inserted in the main text of the manuscript.

5) Please complete the funding information in the online manuscript tracking system. All funding sources mentioned in the Acknowledgments should also be in our system, with project numbers where available.

Completed as requested.

6) Regarding the Author Contributions, we now use CRediT to specify the contributions of each author in the journal submission system. Therefore, please remove the Author Contributions from the manuscript file and make sure that the author contributions in our online manuscript tracking system are correct and up-to-date. The information you specified in the system will be automatically retrieved and typeset into the article. You can enter additional information in the free text box provided, if you wish.

Sentence removed from the main text.

7) Please rename the 'Potential Conflicts of interest' paragraph to 'Disclosure and competing interests statement'. For more information see <https://www.embopress.org/page/journal/14693178/authorguide#conflictsofinterest>

Modified as requested.

8) The main figures appear to have rather low resolution. When I zoom in, the images and text quickly become pixelated. Please provide files that have higher resolution and production quality.

We exported the images as pdf, which seems to keep the appropriate resolution, and also included the powerpoint files with the original figures.

9) Supplementary information file:

A) Figure EV1 to EV5 should be provided as individual production quality figure files. Their legends are part of the main manuscript in a section called Expanded View Figure legends, placed after the main figure legends. The figures need to fit on one page. If needed, you can split one of the figures in 2 figures.

We have now compacted the panels to one page.

B) Please remove the supplementary methods and move them to the main manuscript text.

We have now performed the requested alteration.

C) The remaining Table 1 could be uploaded as Table EV1 (please update the callouts in the manuscript.)

We updated the callout in the manuscript.

10) Callouts that also need correction: Supplementary Table S1, Supplementary Figure 5F

We had now corrected these callouts.

11) Please download and fill our Reagents and Tools Table template (.docx), which you can find in our author guidelines: <https://www.embopress.org/page/journal/14693178/authorguide#structuredmethods>.

An example of a Method paper with Structured Methods can be found here: <https://www.embopress.org/doi/10.15252/msb.20178071>.

We have now remove all the information about reagents from the manuscript and we have now the "Reagent Table" filled with the required information.

12) There is a callout for a Table S2 in the methods section, but such a table is not present. Please check and correct the callout.

We had now corrected to Table EV1.

13) We perform a routine image integrity check on all revised manuscripts. In this case we noticed that the GAPDH blots shown in Figure EV1K and Figure EV3N appear to be the same. Please check these figure panels, clarify and ensure that the appropriate controls are shown for each Western blot/experiment. Are the Western blots shown in these two panels and their controls from the same experiment?

Yes, they are. Both panels are Wt and CLN3-KO cells exposed to 100 nM of BafA1 for 2h from the same samples, running in the same membrane, where both LC3 (for panel EV1K) and pYAP1^{Y357} levels (for panel EV3N) were assessed. Thus, GAPDH is the same in both panels. However, if a different immunoblot or independent experiment is required, we are happy to change one of the panels.

14) Source Data: I did a spot check and the source data for Figure 1D seemed not to match. I could not find the cells shown in the figure panels in the source data image. But this was difficult to assess, due to the limited resolution of the figure files. Please make sure that all source data are correct.

The source file from the Wt cells is correct. However, we have now corrected the source file from CLN3-KO cells.

15) Source Data: In my previous decision letter I told you that we perform a data integrity check on all quantitative source data files and that I had noticed the following aberrations that needed clarification (a-e). I checked and noticed that you have updated the source data as follows (indicated with =>):

a) Source Data xls file for Fig 1K: the values in row 5 (BAX) and row 7 (DR5) are exactly the same. This is also reflected in the figure panel. Please check.
=> You have replaced the data for DR5.

b) Source Data xls file for Fig. 1G: The values for PUMA match exactly with those for DR5 in the .xls file. The values seem also to be the same in the figure panel (bars, error bars and data points are the same).
=> You have replaced the data for DR5.

c) Source data xls file for Fig. 2D: values for Hippocampus and Thalamus are identical. Please check and clarify.

=> You have replaced the Hippocampus data.

d) Source data xls file for Fig.2H: Column B, rows 287 - 323 are repeated as block in Column B, rows 609 - 645. Those two rows of numbers are identical to each other, i.e., B287 = B609, B288 = B610 etc. Please check and clarify.

=> It appears that the entire .xls file and quantification has been replaced as all numbers are different from the previous version.

e) Source data .xls file for Fig. 3E: blocks of measurements seem to have been repeated and duplicated between CLN3-KO and Ima. I attach the color-coded file. If you look at the green numbers, e.g., 1,01192 and then the series of measurements that follows beneath, you will see that the numbers match between CLN3-KO and Ima. If you start again at the green 0,90244 the same pattern emerges with all measurements in the following rows being equal between both conditions. Please check these measurements and please clarify these duplication.

=> The file has been updated, which resulted in a significant change in the violin blot for Ct and Ima shown in Fig. 3E.

I appreciate that you have corrected the discrepant data but could you please add a short note explaining the earlier detected duplications and the changes I noted for Figure 2H and the changes to the quantification results now shown in Figure 3E? Thank you very much.

The errors detected in lines a–c were related to the duplication process of the graph in GraphPad Prism, which was done to ensure a consistent graph style. On the other hand, the error at the lines d and e arose from a copy-and-paste issue caused by differences in the column setup of the Excel file during the normalization process. As a result, some data were inadvertently duplicated, leading to incorrect average values for the WT cells and, consequently, altered normalized values. While most of the dataset remained unaffected and showed the same trend of increase or decrease, the repeated nuclear values were now removed it, changing the violin plot dispersion. In case of Figure 3E, the deletion of this duplication made the differences more evident, as the WT average value was higher than it should be without duplicates.

In Figure 3E of the previous version, this issue is evident in the normalization step: the average of the yellow dots was not equal to 1 for each experiment.

16) Our production/data editors have asked you to clarify several points in the figure legends (see below). Please incorporate these changes in the manuscript and return the revised file with tracked changes with your final manuscript submission.

=> Please note that this analysis was done on the previous version of your

manuscript, but the requests have not been not implemented. The figure panels might have been changed to some extent, so please keep in mind these general specifications: (a) quantifications need information on the number of repeats (technical or biological), the statistical test used, the exact p-values unless $p < 0.0001$, a description of bars and error bars, violin blots or box plots, individual data points need to be added, (b) imaging data need scale bars and any labeling in the image (e.g., arrows or dashed lines) must be defined in the legend (the latter also applies to labels on Western blots).

The requests on the previous version were the following:

A) Figure legend text:

1. Please note that the legends for supplementary figures 1G, H is not provided in the sequential manner (legend for supplementary figure 1H is provided before legend of supplementary figure 1G). This needs to be rectified.

B) Statistical test information. Only p-values that are actually shown in the figure panel(s) should (and must) be defined in the legends, all others should be removed from (or added to) the legend. Moreover, we ask for the specification of exact p-values:

1. Please note that the exact p values are not provided in the legends of figures 1D-H, I-J, L; 2A, B, D, E, F, H, I, J; 3A-F; 4B, C, E-K; supplementary figure(s) 1C, D, H, I, K; 2A, B, C, E, F, H; 3A, C, D, E, F, G, H, J, L, M; 4A, B, C; 5A, C, D, E, G.

We have now provided the exact p-values in the missing panels.

2. Please indicate the statistical test used for data analysis in the legends of figures 1B, C.

We have inserted the information.

3. Please note that in figures 1I, J; supplementary figure(s) 1K, there is a mismatch between the annotated p values in the figure legend and the annotated p values in the figure file that should be corrected.

We hope to have corrected this situation.

C) Replicates and error bars:

1. Please note that the box plots need to be defined in terms of minima, maxima, centre, bounds of box and whiskers, and percentile in the legend of figure 1D.

2. Please note that the error bars are not defined in the legends of figures 1E, F, G, H, I, J, L.

D) Data presentation:

1. Please note that the scale bar needs to be defined for figures 4C, F, H; supplementary figure(s) 3J, 4B, C; 5B, C.
2. Please note that scale bar and its definition are missing for figure 2D.
3. Please note that the yellow dashed border is not defined in the legend of supplementary figures 3G, H, J; This needs to be rectified.

We have now corrected each raised point. Thank you for these annotations.

17) Finally, EMBO Reports papers are accompanied online by

A) a short (1-2 sentences) summary of the findings and their significance,

Loss of CLN3 leads to DNA damage and activation of c-Abl-YAP1-dependent pro-apoptotic signaling which underlies increased apoptosis susceptibility. This pathway provides a novel therapeutic target against Batten disease.

B) 2-3 bullet points highlighting key results and

In conditions of CLN3-loss of function:

- YAP1 signalling is activated and increases the expression of pro-apoptotic gene.
- This pro-apoptotic phenotype is dependent on YAP1 phosphorylation at tyrosine-357 by the tyrosine kinase c-Abl.
- Increased DNA damage contributes to c-Abl activation and subsequent YAP1-mediated cell apoptosis and cell cycle arrest.

C) a schematic summary figure that provides a sketch of the major findings (not a data image).

Please provide the summary figure as a separate file in PNG or JPG format at a size of 550x300-600 pixels (width x height). Please note that the size is rather small and that text needs to be readable at the final size. Please send us this information along with the revised manuscript.

We have removed the panel from Figure 4L from the main figures to be the summary figure asked at this point. We also removed the sentence referring this panel from the main text.

Nuno Raimundo
Penn State College of Medicine
Cellular and Biological Systems
500 University Dr
College Of Medicine, Room C4769
Hershey, PA 17033
United States

Dear Nuno,

I am very pleased to accept your manuscript for publication in the next available issue of EMBO reports. Thank you for your contribution to our journal.

Kind regards,

Martina
